# Severity Benchmarks for the Level of Personality Functioning Scale—Brief Form 2.0 (LPFS-BF 2.0) in Polish Adults

**DOI:** 10.3390/healthcare13030340

**Published:** 2025-02-06

**Authors:** Karolina Juras, Mateusz Mendrok, Janusz Pach, Marcin Moroń

**Affiliations:** 1Faculty of Social Sciences, University of Silesia in Katowice, 40-007 Katowice, Poland; k.juras@o365.us.edu.pl (K.J.); mmendrok@o365.us.edu.pl (M.M.); j.pach@o365.us.edu.pl (J.P.); 2Institute of Psychology, University of Silesia in Katowice, 43-126 Katowice, Poland

**Keywords:** personality impairment, personality disorders, cut-off, psychotherapy seeking, Polish population

## Abstract

**Background/Objectives:** The Level of Personality Functioning Scale—Brief Form 2.0 (LPFS-BF 2.0) is a self-report screening measure of personality impairments according to the DSM-5 Alternative Model for Personality Disorders and the ICD-11 classification of personality disorders. Nevertheless, reliable cut-off scores that could help in clinical decision making are still lacking for many populations. The aim of this study was to develop severity benchmarks of the LPFS-BF 2.0 for a Polish population based on the item response theory (IRT) approach. **Methods:** A sample of Polish adults (*n* = 530) took part in the study. The participants assessed their personality functioning and pathological personality traits and provided information about psychiatric diagnosis and psychotherapy seeking. The severity benchmarks were developed using IRT and validated using mean and frequency comparisons between groups of different personality impairments according to the developed cut-offs. **Results:** Confirmatory factor analysis (CFA) supported a unidimensional model of the LPFS-BF 2.0. The graded IRT model indicated satisfactory item functioning for all LPFS-BF 2.0 items. The normative observed score thresholds at different latent severity levels of personality impairments were developed, and significant overall differences were found between the LPFS-BF 2.0 norm-based severity benchmarks in pathological personality traits and psychotherapy seeking. **Conclusions:** The IRT-based cut-offs for the LPFS-BF 2.0 identified individuals high on pathological personality traits (particularly disinhibition) and were predictive of psychotherapy seeking. The developed severity benchmarks allow for the interpretation of LPFS-BF 2.0 scores, supporting clinical diagnosis and relevant decision making in the Polish population. Practical implications for healthcare practice and research are being discussed.

## 1. Introduction

Personality impairments and disorders cause significant social and personal burden [1]. Their high prevalence [2], associations with poor job performance, risk of deterioration of social roles, suicidality, and comorbidity with other mental disorders [3] make the detection of and intervention in personality impairment a big challenge for healthcare services around the world. Because of shortcomings of the previous categorial approach used to diagnose personality disorders, recent editions of the most important psychiatric manuals (Diagnostic and Statistical Manual of Mental Disorders [DSM-5] and the World Health Organization [WHO] International Classification of Diseases [ICD-11]) proposed trait-based personality pathology models [4]. The Alternative Model for Personality Disorders (AMPD; [5]) introduced in DSM-5 recognizes two criteria of personality disorder: (a) impairments in self and interpersonal functioning (criterion A) and (b) the presence of one or more pathological traits (negative affect, detachment, disinhibition, antagonism, psychoticism) (criterion B). The ICD-11 model of pathological personality traits includes an assessment of personality psychopathology severity (mild, moderate, and severe) and trait domain specifiers, including negative affect, detachment, disinhibition, dissociality (an equivalent of antagonism in DSM-5), and anankastia, but not psychoticism [6]. Despite some differences in the conceptualization of the severity of personality impairment (criterion A) and the role of pathological trait domains in personality disorder diagnosis (criterion B), significant similarity has been shown between trait-based models of personality in ICD-11 and DSM-5 [4,7,8]. A significant difference from the previous categorical approach with arbitrary thresholds for clinically relevant personality disorder pertains to the focus on a dimensional approach to personality functioning. This creates a need for the proper continuous assessment of personality functioning making it possible to differentiate impaired from healthy personality functioning [4].

Although there exist methods with which to assess the pathological personality traits included in criterion B of personality disorders (the Personality Inventory DMS-5; [8,9]), there are fewer measures of self and interpersonal impairment constituting criterion A [5]. The Level of Personality Functioning Scale—Brief Form 2.0 (LPFS-BF 2.0) [10] is a brief and valid measure providing a clinician or psychotherapist with a first impression of a patient’s impairments in personality [11]. The LPFS-BF 2.0 could be a helpful alternative to longer clinical interviews or questionnaires developed to assess criterion A of personality disorders according to the AMPD [12,13,14,15]. Currently, the LPFS-BF 2.0 has been suggested as a part of a standard set of outcome measures in the field of personality disorders by the International Consortium for Health Outcomes Measurement [16]. The LPFS-BF 2.0 developed on the basis of DSM-5 is positively correlated with measures of severity of personality impairment based on ICD-11 [8]. This makes it possible to use the LPFS-BF 2.0 in diagnosis making based on the ICD-11. However, to interpret the severity of personality impairment in the diagnosis of a patient, meaningful thresholds of the LPFS-BF 2.0 should be developed and validated. Establishing the cut-off criteria for the LPFS-BF 2.0 is now necessary to implement this measure appropriately in mental and general health services in individual countries.

Previous studies showed that the LPFS-BF 2.0 reliably measures impairments in identity (experience of oneself as unique, stability of self-esteem, and emotional regulation) and self-direction (self-reflection, pursuit of meaningful goals, and prosocial standards of behavior), as well as impairments in empathy (understanding the influence of one’s behavior on others, tolerance of others’ perspectives, and comprehension and appreciation of others’ experiences; [10]). The first two groups of symptoms assess the level of intrapersonal impairments, while the last two domains assess the level of interpersonal impairments [5]. Together, these four domains represent the “essential commonalities” of personality disorders as defined in DSM-5 [17]. Criterion A of personality disorders reflects the severity of intrapersonal and interpersonal impairments [10]. This severity of personality psychopathology appears to reliably predict current and future psychological functioning [18] and to have concurrent validity against pathological personality traits included in criterion B [19].

The internal structure of the LPFS-BF 2.0 should be unidimensional as intended in criterion A of personality disorders [5]. It was supported in several studies (e.g., [8,12]); however, other studies concluded that the two correlated latent factors structure of the LPFS-BF 2.0 was better supported by empirical data [19,20]. Because criterion A of personality disorders refers to a severity continuum for personality pathology reflecting core impairments in self and interpersonal functioning [21], a unidimensional interpretation of the LPFS-BF 2.0 seems to be better for screening purposes in the initial stages of clinical diagnosis. The validity of the LPFS-BF 2.0 was examined against pathological personality traits (as defined by both DSM-5 and ICD-11), well-being, and symptom severity [10,19,20], as well as against lifetime psychiatric diagnoses, psychotherapies undertaken, and family liability [22]. The LPFS-BF 2.0 has also appeared as sensitivity measure for detecting changes during interventions [10]. The LPFS-BF 2.0 as a brief and valid instrument with satisfying psychometric properties seems to be useful for an initial examination of the severity of personality psychopathology in clinical practice and research. The implementation of the LPFS-BF 2.0 in clinical practice could be based on developing normative scores for the population to screen for individuals with heightened personality pathology [22,23].

### 1.1. Item Response Scaling

Normative values of the LPFS-BF 2.0 were frequently constructed based on the classical test theory (CTT) approaches, which assume a linear relation between a latent variable and test score, equal reliability across scale scores, and a normal distribution as a basis of normalization procedures [24]. Moreover, the focus of CTT is on the test score more than on the item response scaling, and parameters such as reliability, discrimination, and location strongly rely on the sample [24,25]. An alternative approach to the CTT is item response theory (IRT), which models the association between a subject’s score on a latent variable and the probability of a particular response to an item [26]. IRT assumes that the person’s response to an item is a function of their location on a latent variable and of certain characteristics of the respective item [27]. Using information functions for the item and the test score, IRT describes how measurement precision can vary across different levels of the latent variable [28]. It can estimate item properties independently of the sample used [28]; for more details on the differences between CTT and IRT, see, e.g., [29]. In general, IRT offers more precision in the measurement of a person’s latent score and provides more insight into the properties of the items used in the measurement, which are estimated in a non-sample dependent way.

Cut-off scores or severity benchmarks are useful in supporting diagnosis and tracking change during treatment [30]. There are many methods with which to establish meaningful thresholds (e.g., a receiver curve operator [ROC], predictive modeling; see [30,31]). IRT offers a method of determining the cut-off point based on the standard deviation from the latent trait mean [10]. It offers more precision and less dependence on the sample in determining cut-off scores for clinical use. IRT could also better clarify the latent trait range measured by the items and the informative function of response categories used to answer an item [32]. Previous studies showed that IRT-based cut-offs outperform other methods (e.g., ROC or predictive modeling; [30]), especially in cases when the prevalence of the phenomenon tested (here: severe personality impairment) is less than 30% [33], which is the case for personality disorders in the general population [2].

### 1.2. Cultural Invariance of the LPFS-BF 2.0

A recent study on a representative group of the Danish population used IRT to select severity benchmarks for the LPFS-BF 2.0 based on standard deviations from the mean of the latent variable of the LPFS-BF 2.0 total score [11]. It was the first study to suggest cut-off criteria for the LPFS-BF 2.0. Based on the distance from the mean value of the latent variable (θ) representing the severity of personality psychopathology, the suggested score of 26 in the LPFS-BF 2.0 represented the cut-off for a mild level of personality impairments (1.0 standard deviation [SD] above the mean θ, T-score = 60), and a score of 31 reflected a moderate level of personality pathology (1.5 SD above the latent mean; T-score = 65). Likewise, a score of 36 corresponded to a severe level of personality disfunction (2.0 SD above the latent mean; T-score = 70), whereas a score of 41 reflected an extreme level of personality pathology (2.5 SD above the latent mean; T-score = 75). These cut-offs were suggested to be appropriate for Denmark and other Nordic countries, and possibly also for Western countries [11]. However, previous studies had shown that the mean score of the LPFS-BF 2.0 for the Danish population was significantly lower compared to for other European, American, and Asian countries [34]. This could indicate that severity benchmarks for the LPFS-BF 2.0 based on the Danish population could result in false positive detection of personality impairment when used in a different population. Although Natoli et al. [34] showed the measurement invariance of the LPFS-BF 2.0 across tested counties (US, Italy, Denmark, United Arab Emirates), there is no stablished measurement of invariance between the Polish and Danish versions of the LPFS-BF 2.0. Moreover, previous studies on the measurement invariance of the LPFS-BF 2.0 used the confirmatory factor analysis procedure to establish measurement invariance [34,35]. Meanwhile, alternative IRT-based procedures exist and are suggested by the International Test Commission (ITC), which could better test for item invariance between populations [36]. Unfortunately, to our knowledge, there is no study testing the measurement invariance of the IRT model of the LPFS-BF 2.0, which could be important for using IRT-derived cut-offs between populations. In general, population-specific cut-offs, unless they are cross-validated across different regions, might lead to incorrect diagnoses or overdiagnosis, such as false positives, in countries with distinct cultural or social contexts. Thus, further studies verifying the suggested generalizability of the developed cut-offs are needed to improve the implementation of the LPFS-BF 2.0 in public healthcare service in other countries.

### 1.3. The Present Study

In the present study, we used IRT to investigate the severity benchmarks in the Polish version of the LPFS-BF 2.0 to supplement the existing norm values based on the CTT and normal distribution assumptions (T-scores; [22]) with severity benchmarks derived from the distance from the latent mean. First, we examined the Polish version of the LPFS-BF 2.0 using IRT in a sample of Poles matching the general populations in terms of sex and age proportions and tested the validity of the severity benchmarks against pathological personality traits included in ICD-11 (as the manual used in Polish healthcare). We also verified whether psychiatric diagnosis occurrence and seeking psychotherapy differed between individuals, representing various levels of personality psychopathology severity according to the IRT-based cut-offs in the LPFS-BF 2.0. The Polish population was selected because of the initial stage of the research on the validity of the LPFS-BF 2.0 in the Polish population and because of the difficulties in personality impairment and disorder diagnosis in Polish psychiatric healthcare [37]. Polish psychiatric care units tend to focus on disorders from the I axis and lack valid measures that could foster the diagnosis of comorbid personality impairments [22]. Moreover, comprehensive validation of the LPFS-BF 2.0 could warrant its proper use in basic psychiatric screening as recommended by international councils [16].

## 2. Materials and Methods

### 2.1. Participants

In total, 530 adults participated in the study (252 men and 276 women, two identified their sex as different; see Table 1 for a comparison with the general population’s age and sex structure according to Statistics Poland—Główny Urząd Statystyczny [38]). The mean age of the participants was 48.33 (SD = 16.26; range = 18–81). Regarding education level, 1.9% of the participants reported primary education (*n* = 10), lower secondary (*n* = 12; 2.28%), secondary (*n* = 218; 41.45%), post-secondary (*n* = 64; 12.17%), and tertiary (*n* = 228; 43.85%) education. The participants used a 3-point scale ranging from “Below average” to “Above average” to indicate their subjective socioeconomic status. The majority of the participants described their socioeconomic status (SES) as average (*n* = 420; 79.85%), whereas 45 participants identified their SES as lower than average (8.56%) and 61 participants described their SES as higher than average (11.60%). The participants resided mostly in villages (*n* = 207; 39.35%), small cities up to 20,000 inhabitants (*n* = 67), and cities between 20,000 and 99,000 inhabitants (*n* = 102; 19.39%), whereas fewer participants resided in large cities (100,000–500,000 inhabitants; *n* = 85; 16.16%) and the biggest cities (>500,000; *n* = 65; 12.36%). Regarding relationship status at the time of the study, 344 participants were involved in a romantic relationship (65.40%), whereas 182 (34.60%) were not involved in a romantic relationship. Distributions of place of residence and education were similar in other representative samples of the Polish population [39]. The majority of participants were parents of at least one child (*n* = 353; 67.11%). A total of 67 participants had undergone psychotherapy at some point in their life (12.74%) and 43 individuals reported being diagnosed with a psychiatric disorder (8.17%), mostly with depression or anxiety disorders. According to methodological suggestions regarding IRT analysis, the sample size required for the graded response model was higher than 500 [40]. Thus, the number of participants in the present study is appropriate for the IRT analysis of the LPFS-BF 2.0. The only exclusion criterion was age below 18. No particular exclusion criteria regarding mental health were applied.

### 2.2. Measures

The Levels of Psychological Functioning Scale—Brief Form 2.0 (LPFS-BF 2.0; [10]; Polish version: [19]) is a 12-item self-report questionnaire designed to assess impairment in self and interpersonal personality functioning [10]. Each item corresponds to the 12 facets of the LPFS as described in DSM-5 [5]. Respondents rate each item on a scale from 1 (very false or often false) to 4 (very true of often true). Previous studies in Poland corroborated the factorial validity and reliability of the LPFS-BF 2.0 [19]. Norm values were reported as T-scores [22], where scores in a range of 12–20 were described as low, those between 21 and 30 as moderate, and those higher than 30 as severe in terms of the level of personality impairment according to DSM-5 criterion A [5].

The Personality Inventory for ICD-11 (PiCD; [26]; Polish version: [41]) is a 60-item self-report questionnaire assessing five pathological personality traits according to ICD-11: negative affectivity (NA), disinhibition (DN), detachment (DT), anankastia (AK), and dissociality (DL). Each pathological trait is assessed with 12 items rated on a scale from 1 (strongly disagree) to 5 (strongly agree). Previous studies in the Polish population demonstrated factorial validity and high reliability (Cronbach’s α between 0.77 for anankastia and 0.87 for negative affect subscales) of the PiCD [42]. In the present study, the internal consistency for the measured pathological traits was as follows: negative affect: Cronbach’s α = 0.90; disinhibition: Cronbach’s α = 0.91; detachment: Cronbach’s α = 0.88; dissociality: Cronbach’s α = 0.89; and anankastia: Cronbach’s α = 0.80.

### 2.3. Procedure

Data were collected from November to December 2024. The participants were recruited through a nationwide research panel in Poland (https://panelariadna.pl/ (accessed on 24 December 2024)) and invited to complete the questionnaires on an online platform. Participation was voluntary and anonymous. All participants provided informed consent. The participants were informed that they should answer all questions on the online form, resulting in no missing data in the present study. They earned points for their participation in accordance with the panel’s rules. The study was approved by the Institution Review Board of the University of Silesia in Katowice.

### 2.4. Data Analysis

To investigate the data and derive severity benchmarks, we followed the procedure used by Weekers et al. [10] in their study on the Danish general population. First, we used various methods to establish the unidimensionality of the LPFS-BF 2.0. Confirmatory factor analysis (CFA) with a diagonally weighted least squares estimator was chosen because the LPFS-BF 2.0 items are ordered categorically. We used conventional indices to evaluate the unidimensional model fit: root mean square error of approximation (RMSEA), Tucker–Lewis Index (TLI), and Comparative Fit Index (CFI). The cut-off criteria for each index were the following: RMSEA < 0.08 and CFI and TLI > 0.90 [43]. We also inspected individual items’ factor loadings. Additionally, we compared the goodness of fit of the unidimensional structure of the LPFS-BF 2.0 with the two-factor structure (with items 1 to 6 loading on the self-functioning factor and items 6 to 12 loading on the interpersonal functioning factor). We also used parallel analysis and Velincer’s minimum average partial (MAP) test [44]. Then, we estimated the internal consistency of the LPFS-BF 2.0 using MacDonald’s ω. CFA was conducted using R package “lavaan” [45], whereas parallel analysis, the MAP test, and MacDonald’s ω were computed using the “psych” package [46].

Item response theory parameters were estimated using the graded response model (GRM) because of the polytomous response scale [47]. The GRM allows for the estimation of a discrimination parameter for each item and assumes that response options vary on the scale with respect to location only. In the GRM, item difficulties are calculated cumulatively by modeling the probability that an individual will respond to a given response category or higher. Other competing IRT models for polytomous responses include the generalized partial credit model, which models the probability of responding to a specific response category directly (GPCM; [48]). We evaluated whether the GRM and GPCM were distinguishable for the LPFS-BF 2.0, and which model had a better fit using the Vuong test [49,50]. Based on the results of the Vuong test, we decided which model we would use in the analysis. Then, we evaluated the discrimination parameter for each item (a1), as well as the item difficulty parameters (b1–b3), which indicate a particular category k on the response scale how an item reflects the level of the attribute at which patients have a 50% likelihood of scoring a category lower than k versus category k or higher [26]. An item characteristic curve was used to visualize the difficulty parameters of the response scale for each item. We also used the Test Characteristic Curve (TCC), which projects the full test score on the values of the latent variable (theta; θ). We used the TCC to derive the cut-off scores for mild personality impairment (scores between mean θ + 1 SD and mean θ + 1.5 SD), moderate personality impairment (scores between mean θ + 1.5 SD and mean θ +2.0), severe personality impairment (scores between mean θ + 2.0 SD and mean θ + 2.5 SD), and extremely severe personality impairment (scores above mean θ + 2.5 SD). IRT analysis was conducted using R package “mirt” [51].

Lastly, we tested the severity cut-offs of the LPFS-BF 2.0 through ANOVA for pathological personality traits derived from the PiCD. We also used chi-square tests to check the distribution of psychiatric diagnosis and psychotherapy involvement across the severity levels of personality impairment. We predicted the linear trend for pathological personality traits and higher rates of psychiatric diagnosis and psychotherapy seeking in groups with higher severity of personality impairments according to the LPFS-BF 2.0 score (namely, that there will be an increasing number of psychiatric diagnoses and higher rate of psychotherapy seeking depending on the level of personality impairment, e.g., more individuals with moderate impairment will report diagnoses compared to individuals classified as having mild personality impairment, etc.). These analyses were conducted in JASP software 0.17.1 [52].

## 3. Results

The unidimensional structure of the LPFS-BF 2.0 was supported in CFA (χ2 = 54.06; df = 54; *p* = 0.47; RMSEA = 0.001; 90% CI for RMSEA = [0–0.027]; TLI = 1.00; CFI = 1.00). All items had loadings higher than 0.64 on the latent variable. An alternative model with two correlated factors (self and interpersonal functioning) also had good fit to the data (χ2 = 43.68; df = 53; *p* = 0.82; RMSEA < 0.001; 90% CI for RMSEA = [0–0.018]; TLI = 1.00; CFI = 1.00). The two-factor structure better fit the data (Δχ^2^ = 10.38; *p* = 0.001). However, both parallel analysis and MAP analysis indicated one strong dominant dimension (see Figure A1 in Appendix A). Despite the better fit of the two-factor solution in CFA, we found support for a strong dominant dimension, justifying the use of the unidimensional IRT model, which was robust under this condition [53]. McDonald’s ω was 0.92. The mean score for the LPFS-BF 2.0 was 23.36 (SD = 7.08) and did not differ from the mean in previous studies on the LPFS-BF 2.0 conducted in representative sample of Poles [22] (M = 24.08, SD = 7.68; *n* = 1030; t = 1.74; *p* = 0.08; Cohen’s d = 0.10).

Based on the unidimensional structure of the LPFS-BF 2.0, we ran the graded response IRT model, which resulted in a good fit to the data (RMSEA = 0.07; 95% CI for RMSEA = [0.06–0.08]; SRMR = 0.05; TLI = 0.98; CFI = 0.98). An alternative approach, namely GPCM, also had good fit (RMSEA = 0.07; 95% CI for RMSEA = [0.06–0.08]; SRMR = 0.05; TLI = 0.98; CFI = 0.98). However, the Vuong test indicated that these two models were distinguishable (w2 = 0.147; *p* < 0.001) and that the GRM had a better fit compared to the GPCM (z = 6.59; *p* < 0.001). Thus, we proceeded with the GRM IRT model of the LPFS-BF 2.0. The item parameters are given in Table 2. The item characteristic curves are given in Appendix A Figure A1. The item discrimination parameters are all high (>1.77), and difficulty parameters indicate that endorsing response categories 3 and 4 points to significant differences in personality impairment (Table 2).

Figure 1a,b represent the test information curve and the score characteristic curve. The test information curve showed that all reliable scores fall in the theta range of −2.0 to 4.0. Based on the score characteristic curve, the expected score indicating a theta level of θ + 1 SD was 31.91; with θ + 1.5 SD, it was 36.27; with θ + 2 SD, it was 40.38; and with θ + 2.5 SD, it was 43.90 (see Table A1 in Appendix A for expected scores corresponding to various levels of theta). Based on these values, we suggest severity benchmarks for the following: (a) mild personality impairments: LPFS-BF 2.0 total score higher than 32; (b) moderate personality impairment: LPFS-BF 2.0 total score higher than 36; (c) severe personality impairment: LPFS-BF 2.0 total score higher than 40; and (d) extremely severe personality impairment: LPFS-BF 2.0 total score higher than 44.

Next, we divided the participants into four groups depending on the total score of the LPFS-BF 2.0: healthy (<32); mild impairment (32 ≤ score < 36); moderate personality impairment (36 ≤ score < 40); and severe or extremely severe impairment (score ≥ 40). We pooled the last two categories because of the possibly low number of participants representing the extreme level of personality impairment (Table 3).

Next, we conducted ANOVAs for each pathological personality traits included in criterion B of ICD-11 (Table 4). For NA, DT, DN, and DL, we detected a significant effect of the severity of personality impairment (all *p* < 0.001); however, AN did not differentiate regarding the level of personality impairment severity. For NA, post hoc Tukey tests showed that mild, moderate, and severe personality impairment groups did not differ; however, they all differed from healthy participants (all *p* < 0.001). A similar pattern was found for DL and DT. Additionally, there was a marginally significant difference between mild personality impairment and moderate personality impairment groups in DT (*p* = 0.07). Healthy participants reported a lower DN compared to all the other groups (all *p* < 0.001), whereas those with mild level of impairment reported a lower level of DN compared to those with moderate (*p* < 0.01) and severe (*p* < 0.04) personality impairment. The last two groups did not differ. Linear contrasts were significant for DN (t = 6.97, df = 526, *p* < 0.001), DT (t = 6.24, df = 526, *p* < 0.001), DL (t = 4.54, df = 526, *p* < 0.001), and NA (t = 5.96, df = 526, *p* < 0.001), but not for AN (t = −0.18, df = 526, *p* = 0.85). Wilk’s Λ indicates that there were multivariate differences between the groups differentiated with regard to the level of personality impairment (Wilk’s Λ = 0.69; F = 13.63; *p* < 0.001).

There were no sex differences between groups with different severity of personality impairments. Participants involved in a romantic relationship at the moment of the study were equally distributed across the personality impairment severity groups (*p* = 0.92). The healthy group was older compared to the mild (*p* = 0.001) and moderate personality impairment groups (*p* < 0.001). Psychiatric diagnosis was also equally distributed among the severity groups; however, in mild, moderate, and severe personality impairment, participants sought psychotherapy more often compared to healthy individuals (*p* = 0.01).

## 4. Discussion

Personality disorders and personality impairment, in general, are associated with morbidity, higher rates of mortality, and high social and personal costs [54]. The estimated prevalence of personality disorders reaches 10% of the general population [2]. However, according to the dimensional approach to personality psychopathology [5], the higher number of individuals could suffer from subclinical levels of personality impairment [55]. The healthcare system needs measures that can improve screening for personality impairment to enhance clinical decisions and to offer patients the most effective interventions regarding their mental state. However, personality disorder diagnosis yielded difficulties [56] because of the evolving diagnostic criteria in the latest editions of the most important manuals (ICD-11 and DSM-5; [5,6]). The LPFS-BF 2.0 appeared as a well-validated and sensitive measure of core personality impairment according to the newest diagnostic criteria [19]. However, to effectively use it in clinical practice, normative values should be established for a reference population. Although a recent study proposed cut-off criteria for the LPFS-BF 2.0 to distinguish individuals with mild, moderate, and severe personality impairments [8], they were developed using a particular (Danish) population, and therefore, the generalizability of these severity benchmarks should be validated by comparing different populations. The present study used IRT to develop severity benchmarks for the LPFS-BF 2.0 in a Polish sample. To our knowledge, these are the second published severity benchmarks for the LPFS-BF 2.0, and the results show that cut-off criteria may differ depending on the population [11].

The unidimensionality of the total score LPFS-BF 2.0 was supported as the precondition of the further examination of severity benchmarks. Thus, we replicated the results obtained previously by Weekers et al. [11]. Using CFA and IRT, we showed that all items contributed to the total score of the LPFS-BF 2.0, and item functioning was appropriate for all the items. However, we showed that the response options “Sometimes true or somewhat true” and “Very true and often true” had higher difficulty. This is indicated by the high difficulty parameters (b2–b3) for the LPFS-BF 2.0 items (see Table 2). This is in line with previous observations of Weekers et al. [11]. Based on the total score characteristic curve, we developed cut-off criteria at 1, 1.5, 2, and 2.5 SD above the latent mean of personality impairment, similarly to a procedure used in other clinically relevant measures such as the Paranoid Thoughts Scale (see [57]). Our findings show different cut-off scores compared to Weekers et al. [11]. All severity benchmarks were higher compared to the previous study. The first cut-off differentiating people with at least mild personality impairment was 31.91 compared with 25.9 as suggested by Weekers et al. [11]. The second cut-off differentiating individuals with mild personality impairment from those with at least moderate impairment was 36.27 compared with the previously suggested 31. The last two cut-offs differentiating people with severe and extremely severe levels of personality impairment were also higher (4 and 3.5 points, respectively).

First, the cut-offs developed in the present study could be more conservative with regard to the screening function of the LPFS-BF 2.0. However, a closer inspection of the cut-offs developed by Weekers et al. [11] indicates that the crossing threshold between “healthy” personality and mild personality impairment required only two items out of twelve to score 3 (“Sometimes true or somehow true”) and ten items to score 2 (“Sometimes false or somehow false”). In our case, a mild level of impairment required at least eight items to score 3 and the remaining four to score 4 to be screened as being at risk of mild personality impairment. In comparison, cut-offs developed using the Danish sample may be too sensitive (e.g., resulting in false positives) when applied to a different cultural context (e.g., Polish), whereas those developed in our sample may be more conservative in screening for personality impairment in Poland. Second, the differences could result from nation-level differences in personality problems between the Polish and Danish populations. Natoli et al. [34] showed that Danish participants reported the lowest latent mean of the LPFS-BF 2.0 across seven countries (Canada, Chile, Denmark, Germany, Italy, United States of America, and United Arab Emirates). Thus, we believe that the generalizability of the LPFS-BF 2.0 cut-offs developed using the Danish sample may be in fact too sensitive and increase the rate of false positive diagnoses of personality impairment in other populations. In particular, using the cut-off score derived from the Danish population may not be proper because of the lack of established measurement invariance between the Polish and Danish version of the LPFS-BF 2.0.

We examined the validity of the severity benchmarks developed using severity group comparisons with regard to pathological personality traits as described in criterion B of the ICD-11, psychiatric diagnoses, and psychotherapy seeking. We showed that the first severity benchmark (between healthy personality functioning and mild personality impairment) differentiated between individuals with lower personality impairment severity (in the healthy personality group) and any personality impairment severity level in all but one pathological personality trait. Only anankastia did not differentiate between the personality impairment groups. One possible explanation for this observation is that anankastia is not conceptualized directly in the APDM from DSM-5. Since the LPFS-BF 2.0 is based on the APDM, the difference in the conceptualization of pathological personality traits might affect the sensitivity of the LPFS-BF 2.0 to screen for pathological levels of anankastia. However, previous studies showed that measures of personality impairment severity developed for the ICD-11 were highly positively correlated with the LPFS-BF 2.0 [8]. Thus, there could be alternative reasons underlying the lack of sensitivity of severity benchmarks for the LPFS-BF 2.0 in screening for anankastia. For example, social norms could affect social desirability in responding to anankastia more strongly compared to other pathological traits. Thus, individuals will assess their anankastic tendencies as higher despite the level of their personality impairment. The issue of the reliable and valid identification of serious personality impairment associated with rigidity and anankastia should be addressed in future studies with measures for the general severity of personality impairment. Despite the problems with the identification of individuals high on anankastia, other pathological traits, disinhibition, detachment, dissociality, and negative affect, were higher among people classified as having mild, moderate, and severe personality impairment according to the developed severity benchmarks. Moreover, individuals classified as having mild personality impairment differed from those with moderate and severe personality impairment in disinhibition, and marginally in detachment. Our results are similar to those obtained in other studies on the association between the level of personality impairment and indices of mental problems [58]. This observation indicates that screening for mild personality impairment could be important, as from this level of impairment, we observe a notable deterioration in mental well-being. From the healthcare point of view, primary care could benefit from screening for people even with mild personality impairment to offer them some interventions as a preventive measure against the possible future deterioration of their functioning. However, the second cut-off was also sensitive in screening for individuals with higher impulsivity and isolation. Thus, it could be helpful in detecting people with possible dominant problems in self-control and intimacy. It could also drive early interventions targeting the predominant problems of individuals screened as moderately or severely impaired in terms of personality functioning. In general, there were linear positive trends in pathological personality traits and severity level according to the developed cut-offs. Anankastia was an exception; thus, the severity benchmarks for the LPFS-BF 2.0 could be less sensitive to obsessive-compulsive personality pathology.

We also noticed that the cut-offs developed for the LPFS-BF 2.0 were positively associated with individuals’ actual intentions to undertake psychotherapy. Individuals classified as at risk of mild, moderate, or severe personality impairment reported higher frequency of psychotherapy seeking compared to those who were classified as “healthy”. Thus, addressing psychotherapy or other equivalent interventions according to the developed severity benchmarks could be valid. However, we did not find positive correlations with psychiatric diagnoses. The majority of diagnoses in our study referred to unipolar disorders or anxiety disorders comorbid with personality disorders, at around 45% and 25%, respectively [3,59]. However, in our study, the participants did not have a psychiatric diagnosis of personality disorder; thus, the comorbidity of psychiatric diagnosis prevalence [3,59] could have limited comparability to our results. Moreover, we found that psychiatric diagnosis in the mild (16%) and moderate (13%) severity of personality impairment was more frequent, albeit not statistically significant, compared to “healthy” individuals (7.5%). Future studies should verify our cut-offs by combining them with diagnoses of personality disorders. However, difficulties in diagnosing personality disorders [56] could result in some mismatch between the severity levels of personality functioning and formal diagnosis of these disorders. The developed severity benchmarks could help in reducing this gap through the use of well-validated screening measures to support the diagnostic process.

### 4.1. Limitations

In the present study, we developed severity benchmarks for the application of the LPFS-BF 2.0 in clinical practice in Poland. We based our cut-off scores on a sample of Poles representing the general population in terms of the proportions of sex and age and on the IRT-based distances from the latent mean score. The proposed severity benchmarks were more conservative in screening for personality impairments compared to the cut-offs developed by Weekers et al. [11], possibly omitting false positive categorizations. The differences in the suggested cut-off points, particularly for mild impairment could result from a lack of cultural invariance of the LPFS-BF 2.0 between the Polish and Danish populations. Possible sources of such differences could be sought in the differences in responding styles, differences in social desirability or social norms underlying the responses, or different reference points when making statement about oneself in the two populations [60]. When applied directly to the Polish population, Weekers et al.’s [11] severity benchmarks could indicate that individuals without personality impairments would be classified as mildly or moderately impaired in the Polish context. Thus, deriving the cut-off points based on the Polish sample could make it possible to overcome the errors connected with applying severity benchmarks from another culture without testing measurement invariance as a precondition of such application.

However, our study had some limitations that should be addressed in future studies. First, in the studied sample, there were no cases of reported personality disorders. Thus, future studies should compare clinical and non-clinical samples examining whether individuals with a diagnosed personality disorder have higher scores on the LPFS-BF 2.0. Second, the LPFS-BF 2.0 is a self-report measure that could be nonsensitive for people with reduced awareness of their personality impairments [11]. Thus, combining assessments from clinicians or other information sources with patients’ self-reports could help in establishing the sensitivity of the LPFS-BF 2.0 in all manifestations of personality impairments. Third, more measures of well-being and mental health could be used to inspect the validity of the cut-off criteria. Moreover, a higher number of participants could ensure statistical power to detect differences between the groups differentiated based on the severity of personality impairment diagnosed with the LPFS-BF 2.0. In the present study, we warranted the sample size necessary for IRT analysis; however, the number of participants in groups with moderate severity and high severity appeared to be limited. This could affect the power to detect the differences between those groups, on the one hand, and mildly impaired and “healthy” individuals, on the other hand. However, the linear contrast analysis showed a good fit to the data, indicating a positive association between the level of personality impairment and the severity of the criterion variables. The developed cut-off point should also be verified using comparisons with clinical populations (e.g., patients with confirmed diagnosis of personality disorders). Future studies should also test the usefulness of developing independent cut-off scores for two subdimensions of the total LPFS-BF 2.0 score. Here, we conclude that the LPFS-BF 2.0 has one strong dominant dimension using parallel analysis, the MAP test, and the fit of the unidimensional CFA. However, the alternative two-factor model of the LPFS-BF 2.0 also had a good fit to the data, and future IRT analyses could focus on both dimensions exclusively. Developing two instead of one cut-off score for the LPFS-BF 2.0 would be in line with the DSM-5 formulation of decision making in criterion A of personality disorders, where the clinician rating of at least one personality functioning domain as moderately impaired or higher is required for personality disorder diagnosis [20]. Future studies could use more community-based participants (e.g., from primary care facilities) to validate the LPFS-BF 2.0 cut-offs derived from the online research panel in a more ecologically valid sample of users of public healthcare. Moreover, the participants could report on lifetime romantic involvement to precisely establish the association between personality impairment and problems with entering into romantic relationships. Lastly, future studies should test the measurement invariance of the LPFS-BF 2.0 in Polish and Danish population to better explain the reasons for the differences in the cut-off scores obtained (e.g., the role of cultural context).

### 4.2. Implications for Healthcare Practice

Personality impairments and disorders cause important societal costs including higher work absenteeism, difficulties in childcare, a higher rate of psychiatric disorders such as mood disorders, a worse prognosis for the treatment of other mental health disorders, and a higher risk of suicide [1,3,22,41]. Mental health services in many countries lack a unified and efficient system of screening for personality impairment in patients seeking psychotherapy or psychiatric consultation [61]. A brief and valid measure of personality impairment with well-validated cut-offs could improve the process of clinical diagnosis and intervention selection. A measure such as the LPFS-BF 2.0 meets these criteria. Thus, the LPFS-BF 2.0 was recommended as a part of a standard set of outcome measures in the field of personality disorders by the International Consortium for Health Outcomes Measurement [16]. The present study enhances the implementation of the LPFS-BF 2.0 for Polish mental healthcare units and services by developing validated severity benchmarks to detect mild, moderate, and severe personality impairment. The LPFS-BF 2.0 could be used to improve psychotherapeutic diagnosis and during the psychiatric diagnosis process. For example, the LPFS-BF 2.0 could be incorporated into the model of diagnosis of personality disorders or in the general standards of psychiatric diagnosis to improve the screening for personality psychopathology which is neglected in Polish healthcare [37]. However, the developed severity benchmarks should be previously consulted with experts in personality disorder diagnosis to further validate them. They should be cross-validated with clinician’s judgment based on the in-depth diagnosis of personality pathology. Based on screening for personality impairment, clinicians or psychotherapists could include additional interventions or techniques that could address their patient’s personality impairments (pharmacological interventions, psychodynamic or schema therapy, assistance of psychiatric nurses) [37]. However, using the LPFS-BF 2.0 as an additional screening tool may result in some difficulties. For example, it could result in more comorbidity in diagnosis. This could require the development of new recommendations on how to prepare a treatment protocol in case of comorbid personality impairment. In general, such changes in healthcare organization could result in a more detailed and individualized approach to the patient’s psychopathology and in the development of more effective care. It could also be used in personnel selection in a particular instance (e.g., military, special forces recruitment). Future studies could examine its utility in forensic psychology and psychiatry, as well as, in general, healthcare (e.g., to aid medical treatment and consultations).

### 4.3. Implications for Research

The present study supports the usefulness of IRT to derive severity benchmarks for clinical measures [32]. IRT analysis of the LPFS-BF 2.0 items could, moreover, indicate how precise all items are in mapping the range of the latent trait of personality impairment and how to interpret the test taker’s responses. For example, the present study showed that test scores of the LPFS-BF 2.0 reflect the theta range from −2 to 3, whereas for Weekers et al. [11], the theta range was narrower (0 to 3). This observation could have consequences for understanding the latent variable measured with the LPFS-BF 2.0. The present study suggests that low scores of the LPFS-BF 2.0 may represent the opposite pole (healthy personality functioning) with regard to severe personality impairment linked with high scores on the LPFS-BF 2.0. However, in other populations, the LPFS-BF 2.0 seems to map the lack of or intensity of personality impairment only. Another important goal of future studies is to establish measurement invariance for the LPFS-BF 2.0 using IRT analysis [36]. Moreover, the IRT-derived model of measuring personality impairment using the LPFS-BF 2.0 makes it possible to overcome the limitations of conventional, CTT-based application of the LPFS-BF 2.0 (e.g., sample dependence).

## 5. Conclusions

The severity benchmarks for the LPFS-BF 2.0 developed in the present study are a second set of normative scores proposed to enhance clinical decisions in the diagnosis of personality impairments. Compared to the cut-off derived from the Danish population [11], they are higher and, thus, more conservative. However, the higher cut-offs obtained in the present study could reflect the lowered mean personality impairment among Danish citizens compared to in other European, American, and Asian countries, as shown in previous studies [34]. Thus, the present severity benchmarks seem to be more appropriate for countries with a higher national-level mean of LPFS-BF 2.0 scores such as Italy or Germany. The developed cut-offs could help in detecting people with mild, moderate, and severe levels of personality impairment, potentially improving early detection and intervention and reducing the societal and personal burden of personality impairments.

## Figures and Tables

**Figure 1 healthcare-13-00340-f001:**
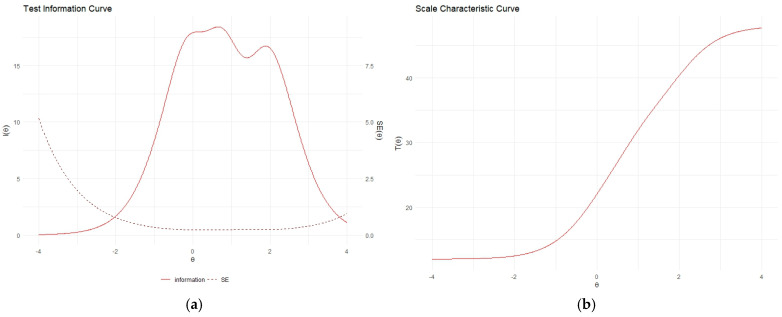
Test information curve (**a**) and scale characteristic curve (**b**).

**Table 1 healthcare-13-00340-t001:** Sex and age structure of the participants of the present study and of the general population in Poland.

	Population		The Present Study	
Variable	*n*	%	*n*	%
Total	30,771,627		530	
Sex				
Men	14,659,525	47.64	252	47.55
Women	16,112,102	52.36	276	52.08
Other			2	0.004
Age Group				
18–29	4,580,625	14.89	77	14.53
29–39	5,564,766	18.08	114	21.51
40–49	6,107,139	19.85	95	17.92
50–59	4,625,376	15.03	89	16.79
60–69	4,863,417	15.80	102	19.25
70+	5,030,304	16.35	58	10.94

**Table 2 healthcare-13-00340-t002:** Item loadings, item discrimination, and item difficulty parameters for the LPFS-BF 2.0.

Item	Factor Loading	a1	b1	b2	b3
LPFS1	0.78	3.05	0.01	0.77	1.71
LPFS2	0.68	2.00	−0.56	0.62	1.98
LPFS3	0.79	3.08	−0.14	0.85	1.92
LPFS4	0.71	2.29	−0.24	0.69	1.87
LPFS5	0.78	2.96	−0.26	0.66	1.97
LPFS6	0.70	2.13	−0.47	0.69	2.25
LPFS7	0.64	1.77	−0.96	0.45	2.33
LPFS8	0.66	1.91	−0.50	0.81	2.53
LPFS9	0.68	1.98	−0.74	0.67	2.38
LPFS10	0.64	1.96	0.17	1.18	2.45
LPFS11	0.75	2.62	−0.08	0.84	2.20
LPFS12	0.70	2.09	−0.56	0.73	2.33

**Table 3 healthcare-13-00340-t003:** Severity benchmarks for personality impairment measured by total score of the LPFS-BF 2.0.

Severity Benchmarks	Weekers et al., 2023 [11]	The Present Study	Suggested Cut-Off
Mean θ + 1 SD (84th percentile)	25.9	31.91	32
Mean θ + 1.5 SD (92nd percentile)	31	36.27	36
Mean θ + 2.0 SD (97th percentile)	36	40.38	40
Mean θ + 2.5 SD (99th percentile)	40.5	43.90	44

**Table 4 healthcare-13-00340-t004:** Means, standard deviations, and group comparisons for personality impairment severity groups.

Variable	Healthy (*n* = 442)		Mild (*n* = 44)		Moderate (*n* = 32)		Severe (*n* = 12)			
	M	SD	M	SD	M	SD	M	SD	F	η^2^
NA	2.70	0.65	3.45	0.50	3.64	0.58	3.78	0.52	46.46	0.21
DT	2.52	0.63	3.09	0.52	3.44	0.46	3.55	0.52	41.18	0.19
DL	2.15	0.62	2.68	0.70	3.00	0.69	2.90	0.81	29.12	0.14
DN	2.12	0.62	2.74	0.68	3.20	0.66	3.29	0.77	50.62	0.22
AN	3.40	0.50	3.49	0.42	3.41	0.46	3.40	0.50	0.49	0.01
Age	50.06	15.99	40.73	15.64	36.34	13.34	39.83	12.70	12.53	0.07
	*n*	%	*n*	%	*n*	%	*n*	%	χ^2^	V
Sex (women)	234	52.94	23	52.27	15	46.88	4	33.33	6.88	0.08
Romantic relationship (yes)	289	65.39	29	65.91	19	59.38	8	66.67	0.50	0.03
Psychiatric diagnosis (yes)	32	7.24	7	15.91	4	12.50	1	8.33	4.74	0.10
Psychotherapy seeking (yes)	48	10.86	10	22.73	8	25.00	3	25.00	11.06	0.14

## Data Availability

Data for the analysis are available at https://osf.io/jnw5k/?view_only=872dc666d0084f1b9b5b0225a201c3f9 (accessed on 24 December 2024).

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
