# Peer review of "Severity Benchmarks for the Level of Personality Functioning Scale—Brief Form 2.0 (LPFS-BF 2.0) in Polish Adults"

_healthcare, 2025, doi:10.3390/healthcare13030340_

Round 1

Reviewer 1 Report

Comments and Suggestions for Authors

The manuscript by Juras et al. presents the results of a psychometric study investigating the dimensional and measurement properties of the Level of Personality Functioning Scale – Brief Form (LPFS-BF) in a sample of Polish adults using item response theory methods. Overall, the paper is well-structured, well-referenced, and employs sound statistical methodology. The topic —assessing nuances of personality functioning using dimensional models, such as the Alternative Model for Personality Disorders (AMPD) from DSM-5— is highly relevant, and studies evaluating feasible measurement instruments are warranted.

In my judgment, the study merits consideration for publication, with only minor revisions suggested. Below, I provide my section-by-section, point-by-point observations and comments for the authors, highlighting specific aspects that could be addressed during the revision process.

[Introduction]
The introductory section effectively outlines the background and provides appropriate references to the current literature. However, I noticed a likely typo in lines 72–73. If I understand correctly, the intended phrase is: "The internal structure of the LPFS-BF 2.0 should be unidimensional […]".

[Materials and Methdos]
Several observations:

- Line 128: The acronym "GUS" is not introduced before its first usage.
- How was socioeconomic status assessed? What reference or scale guided participants’ self-assessments as "average," "lower than average," etc.?
- In Table 1, the captions for the two columns ("Population" and "The present study") appear to be inverted.
- The authors' effort to compare sex and age group proportions with official statistics from the general population is commendable. However, the claim that the sample is representative could be tempered, considering the sample size and the recruitment procedure.
- In the "Data Analysis" subsection:
    - References for the reported CFA index cutoffs should be included.
    - A brief explanation of the rationale behind the benchmark identification procedure could be introduced here.
    - Given the advanced statistical methods employed, specifying the software(s) used for analyses would enhance replicability.
- Line 200: The phrase "We predicted the linear trend for pathological personality traits […]" is unclear.

[Results]
This section is straightforward, with results clearly presented. I was wondering whether the authors considered testing alternative CFA models to further evaluate the unidimensional structure of the LPFS-BF. As it stands, the authors have evaluated only a single-factor model (presumably with all items loading on a single factor and no error covariances) with reference to conventional cut-offs for fit indices. Testing competitive models (e.g., the two-factor self-functioning and interpersonal functioning model tested by Weekers et al.) might have provided additional support for unidimensionality.

[Discussion]
I appreciated the authors’ thoughtful reasoning and critical appraisal of their findings in the context of the existing literature. The strengths, implications, and limitations of the study are thoroughly and clearly discussed. I have two minor comments:

- The procedure for establishing cutoff criteria for the latent score of personality impairment is reported to have been previously used for the Minnesota Multiphasic Personality Inventory (MMPI) (lines 288–291). However, the cited reference (reference number 32, Cella et al., 2014) appears to pertain to the PROMIS item bank and does not mention the MMPI. While a similar procedure for scoring severity is indeed used for the MMPI, the reference seems not proper.
- Regarding the lack of differences in personality impairment at varying levels of anankastia, did the authors consider that the LPFS-BF was likely developed based on the AMPD model? Unlike the ICD framework, the AMPD does not conceptualize pathological personality traits to include anankastia. If relevant, this aspect could be discussed further to highlight the importance of the conceptual model of reference (and its subtle nuances) in shaping the operational meaning of personality impairment and pathology.

Author Response

The Response to the Reviewer #1

Dear Reviewer 1,

Thank you for all insightful suggestions to the previous version of the manuscript. Below, we showed how we addressed your suggestions. Where necessary, we included line numbers from the clean version of the manuscript. These changes are also highlighted in the second uploaded file with changes tracked.

We hope that the revised version of the manuscript addresses your reservations to the previous version of the manuscript.

Thank you for all comments and efforts to improve this submission,

The The Authors.

Reviewer #1: The manuscript by Juras et al. presents the results of a psychometric study investigating the dimensional and measurement properties of the Level of Personality Functioning Scale – Brief Form (LPFS-BF) in a sample of Polish adults using item response theory methods. Overall, the paper is well-structured, well-referenced, and employs sound statistical methodology. The topic —assessing nuances of personality functioning using dimensional models, such as the Alternative Model for Personality Disorders (AMPD) from DSM-5— is highly relevant, and studies evaluating feasible measurement instruments are warranted.
In my judgment, the study merits consideration for publication, with only minor revisions suggested. Below, I provide my section-by-section, point-by-point observations and comments for the The Authors, highlighting specific aspects that could be addressed during the revision process.
The Authors: Thank you for pointing out the strengths of the manuscript.

Reviewer #1: [Introduction]
The introductory section effectively outlines the background and provides appropriate references to the current literature. However, I noticed a likely typo in lines 72–73. If I understand correctly, the intended phrase is: "The internal structure of the LPFS-BF 2.0 should be unidimensional […]".

The Authors: Thank you for noticing this typo. We have corrected it. Please, see

Reviewer #1: [Materials and Methdos]
Several observations:
- Line 128: The acronym "GUS" is not introduced before its first usage.

The Authors: We have now expanded the acronym: Statistics Poland – GÅ‚ówny UrzÄ…d Statystyczny.

Reviewer #1:- How was socioeconomic status assessed? What reference or scale guided participants’ self-assessments as "average," "lower than average," etc.?

The Authors: We have clarified this issue. The participants used the 3-point scale ranging from “Below average” to “Above average” to indicate their subjective socioeconomic status (ll. 190-191 in clean version).

Reviewer #1:- In Table 1, the captions for the two columns ("Population" and "The present study") appear to be inverted.

The Authors: Thank you for noticing this issue. We reversed the names of the columns.

Reviewer #1:- The The Authors' effort to compare sex and age group proportions with official statistics from the general population is commendable. However, the claim that the sample is representative could be tempered, considering the sample size and the recruitment procedure.

The Authors: We deleted the “representative” from the abstract (l.16 of the previous version) and in Discussion (l.281 of the previous version). Additionally, we changed two sentences by using: “a sample of Poles representing the general population in terms of the proportions of sex and age” instead of a representative sample (l. 169-170; l. 479-480).

Reviewer #1:- In the "Data Analysis" subsection:
    - References for the reported CFA index cutoffs should be included.

The Authors: We have added the reference (43. Hu, L.-T.; Bentler, P. M. Cutoff criteria for fit indexes in covariance structure analysis: Conventional criteria versus new alternatives. Stru Equ Modeling 1999, 6(1), 1–55. https://doi.org/10.1080/10705519909540118).

Reviewer #1:    - A brief explanation of the rationale behind the benchmark identification procedure could be introduced here.
The Authors: The rationale was added in lines 70-74 (clean version):

However, to interpret the severity of personality impairment in the diagnosis of a patient meaningful thresholds of the LPFS-BF 2.0 should be developed and validated. Establishing the cut-off criteria for the LPFS-BF 2.0 is now necessary to implement this measure appropriately in mental and general health services in the individual countries.

And in lines 121-131 (clean version):

Cut-off scores or severity benchmarks are useful in supporting diagnosis and tracking change during treatment [30]. There are many methods to establish meaningful thresholds (e.g., a receiver curve operator [ROC], predictive modeling; see 30-31). IRT offers a method of determining the cut-off point based on the standard deviation from the latent trait mean [10]. It offers more precision and less dependence on the sample in determining cut-off scores for clinical use. IRT could also better clarify the latent trait range measured by the items and the informative function of response categories used to answer an item [32]. Previous studies showed that IRT-based cut-offs outperform other methods (e.g, ROC or predictive modeling; 30), especially in cases when the prevalence of the phenomenon tested (here: severe personality impairment) is less than .30 [33] which is the case for personality disorders in the general population [2].

Reviewer #1:    - Given the advanced statistical methods employed, specifying the software(s) used for analyses would enhance replicability.

The Authors: We have added the following sentences to the Data analysis section:

  • 254-256: CFA was conducted using R package “lavaan” [45], whereas parallel analysis, the MAP test, and MacDonald’s ω were computed using the “psych” package [46].
  • 277-278: The IRT analysis was conducted using R package “mirt” [51].
  • 288-289: These analyses were conducted in JASP software [33].

Reviewer #1: - Line 200: The phrase "We predicted the linear trend for pathological personality traits […]" is unclear.

The Authors: We addd the following clarification in the parenthsis:

  • 282-288: We predicted the linear trend for pathological personality traits and higher rates of psychiatric diagnosis and psychotherapy seeking in groups with higher severity of personality impairments according to the LPFS-BF 2.0 score (namely, that there will be increasing number of psychiatric diagnosis and psychotherapy seeking depending on the level of personality impairment, e.g., more individuals with moderate impairment will report diagnosis compared to individuals classified as having mild personality impairment, etc.)

Reviewer #1: [Results]
This section is straightforward, with results clearly presented. I was wondering whether the The Authors considered testing alternative CFA models to further evaluate the unidimensional structure of the LPFS-BF. As it stands, the The Authors have evaluated only a single-factor model (presumably with all items loading on a single factor and no error covariances) with reference to conventional cut-offs for fit indices. Testing competitive models (e.g., the two-factor self-functioning and interpersonal functioning model tested by Weekers et al.) might have provided additional support for unidimensionality.

The Authors: Thank you for this suggestion. We added more criteria to test the unidimensionality of the LPFS-BF 2.0 for IRT analysis (see Data analysis section):

  • 242-253: To investigate the data and derive severity benchmarks, we followed the procedure used by Weekers et al. [10] in their study in the Danish general population. First, we used various methods to stablish unidimensionality of the LPFS-BF 2.0. Confirmatory factor analysis (CFA) with a diagonally weighted least squares estimator was. chosen because the LPFS-BF 2.0 items are ordered categorically. We used conventional indices to evaluate unidimensional model fit: root mean square error of approximation (RMSEA), Tucker-Lewis Index (TLI), and Comparative Fit Index (CFI). The cut-off criteria for each index were: RMSEA < .08, CFI and TLI > .90 [43]. We also inspected individual items’ factor loadings. Additionally, we compared the goodness of fit of the unidimensional structure of the LPFS-BF 2.0 with the two-factor structure (with items 1 to 6 loading on the self-functioning factor, and items 6 to 12 loading on the interpersonal functioning factor). We also used parallel analysis and Velincer’s minimum average partial (MAP) test [44].

In the Results section we add more nuanced results on the dimensionality of the LPFS-BF 2.0:

  • 291-300: The unidimensional structure of the LPFS-BF 2.0 was supported in CFA (χ2 = 54.06; df = 54; p = 0.47; RMSEA = .001; 90% CI for RMSEA = [0 – .027]; TLI = 1.00; CFI = 1.00). All items had loadings higher than .64 on the latent variable. An alternative model with two correlated factors (self- and interpersonal functioning) also had good fit to the data (χ2 = 43.68; df = 53; p = 0.82; RMSEA < .001; 90% CI for RMSEA = [0 – .018]; TLI = 1.00; CFI = 1.00). The two-factor structure had better fit to the data (Δχ2 = 10.38; p = .001). However, both parallel analysis and MAP analysis indicated one strong dominant dimension (see Figure A1 in the Appendix). Despite the better fit of the two-factor solution in CFA, we found support for a strong dominant dimension, justifying the use of the unidimensional IRT model which was robust under this condition [53].

Based on all these results, we decided to proceed with unidimensional model of the LPFS-BF 2.0. However, we added the following limitation to the Limitations section:

  • 514-522: Future studies should also test the usefulness of developing independent cut-off scores for two subdimensions of the total LPFS-BF 2.0 score. Here, we concluded that the LPFS-BF 2.0 has one strong dominant dimension using parallel analysis, the MAP test and fit of unidimensional CFA. However, the alternative two-factor model of the LPFS-BF 2.0 also had good fit to data and future IRT analyses could focus on both dimensions exclusively. Developing two instead of one cut-off score for the LPFS-BF 2.0 would be in line with the DSM-5 formulation of decision making in criterion A of personality disorders, where clinician rating of at least one personality functioning domain as moderately impaired or higher is required for personality disorder diagnosis [20].

Reviewer #1: [Discussion]
I appreciated the The Authors’ thoughtful reasoning and critical appraisal of their findings in the context of the existing literature. The strengths, implications, and limitations of the study are thoroughly and clearly discussed. I have two minor comments:

- The procedure for establishing cutoff criteria for the latent score of personality impairment is reported to have been previously used for the Minnesota Multiphasic Personality Inventory (MMPI) (lines 288–291). However, the cited reference (reference number 32, Cella et al., 2014) appears to pertain to the PROMIS item bank and does not mention the MMPI. While a similar procedure for scoring severity is indeed used for the MMPI, the reference seems not proper.

The Authors: We deleted the reference. Instead, we added a reference to:

  1. 386-389: Based on the total score characteristic curve, we developed cut-off criteria at 1, 1.5, 2, and 2.5 SD above the latent mean of personality impairment similar to a procedure used in other clinically-relevant measures such as the Paranoid Thoughts Scale (see [57]).
  2. Freeman, D.; Loe, B. S.; Kingdon, D.; Startup, H.; Molodynski, A.; Rosebrock, L.; Brown, P.; Sheaves, B.; Waite, F.; Bird, J. C. The revised Green et al., Paranoid Thoughts Scale (R-GPTS): psychometric properties, severity ranges, and clinical cut-offs. Psychol Med 2021, 51(2), 244–253. https://doi.org/10.1017/S0033291719003155

Reviewer #1:- Regarding the lack of differences in personality impairment at varying levels of anankastia, did the The Authors consider that the LPFS-BF was likely developed based on the AMPD model? Unlike the ICD framework, the AMPD does not conceptualize pathological personality traits to include anankastia. If relevant, this aspect could be discussed further to highlight the importance of the conceptual model of reference (and its subtle nuances) in shaping the operational meaning of personality impairment and pathology.

The Authors: Thank you for this suggestion. We addressed this issue in lines 424-437:

One possible explanation of this observation is that Anankastia is not conceptualized directly in the APDM from DSM-5. Since the LPFS-BF 2.0 is based on the APDM, the difference in conceptualization of pathological personality traits might affect the sensitivity of the LPFS-BF 2.0 to screen for pathological levels of Anankastia. However, previous studies showed that measures of personality impairment severity developed for the ICD-11 were highly positively correlated with the LPFS-BF 2.0 [8]. Thus, there could be alternative reasons underlying the lack of sensitivity of severity benchmarks for the LPFS-BF 2.0 in screening for Anankastia. For example, social norms could affect social desirability in responding to Anankastia more strongly compared to other pathological traits. Thus, individuals will assess their anankastic tendencies as higher despite the level of their personality impairment. The issue of reliable and valid identification of serious personality impairment associated with rigidity and anankastia should be addressed in future studies with measures of general severity of the personality impairment.

Reviewer 2 Report

Comments and Suggestions for Authors

Abstract.
The Abstract provides an excellent summary of your research manuscript.  A few writing suggestions include the following.
1. replace ". . . assessed their personality 'impairments'" with ". . . assessed their personality 'functioning'"
2. avoid writing in the first person, e.g., "We developed . . ."
3. ". . .implications for healthcare are being discussed."  ". . .implications for healthcare practice and research are discussed."

Keywords.
1. Consider keywords specific to the manuscript as well as keywords provided for the Weekers et al. (2023) publication; e.g., Polish population,
2. Item Response Theory, validity are irrelevant keywords (note my comment regarding the title.

Title.
1. item response theory is merely the most appropriate modeling framework for completing the analyses, and does not merit emphasis in the title unless the method is somehow more than 'the method'. For this research publication, in my opinion mention of Item Response Theory in the title is somewhat distracting.

Introduction.
Overall. The introduction includes all substantive content. A few writing corrections are recommended.  Note a complex suggestion in item 3 below suggesting re-organization of key content.
1. writing. page 2, line 72. ". . . LPFS-BF 2.0 should the unidimensional as intended . . ."  (should be unidimensional)
2. page 2, line 86. Consider subheading paragraph 4 'Item Response Scaling'.  This is a central idea to the manuscript you want to provide as you emphasize the scaling technology.
3. Consider re-organizing the content of paragraphs 4 (page 2, line 86) and paragraph 5 (page 3, line 113).  The content of one paragraph is 'Item Response Scaling' (paragraph 3). The content of the next paragraph 4 is 'Lack of Cultural Invariance' in which you explain the need for a Polish scaling given that the Danish scaling is inappropriate, i.e., scaling is not invariant.  
4. Note, pertaining to lack of cultural invariance you may consider reference to the International Test Commission (ITC) https://www.intestcom.org/ guidelines and concepts/technologies, including IRT pertaining to scaling measures across populations from different cultures/languages.

Materials and Methods.
Overall. The methods are appropriate, though a few suggestions are provided.
1. Table 1.  The column titles need to be reversed. The population is the first column, the 'present study' is the second column.
2. Measures. For the PiCD, report the reliability coefficients and not only the references for those psychometrics.
3. page 5, paragraph 2. Your description of IRT modeling and model choice (graded response) needs revision. First, consider a more complete explanation of why the 'graded response model'.  Rather than merely ". . .because of the polytomous response scale", include the model constraint that response options vary on scale with respect to location only, i.e., they are equally sensitive.  
4. Preceding note 2 above points out a serious error in your writing. Page 5, lines 189-190. You state, "Item characteristic curve was used to visualize the discriminatory properties of the response scale for each item."  NOTE.  Each item has a different 'difficulty' parameter, not discrimination parameter, discrimination is constant across items within the graded response model.  

Results.
Overall. The results are complete, but need some revision, particularly correction to Table 2 title.
1. Refer to comment 4 above regarding graded response model parameters.  Table 2 title has parameter identification reversed, 'a1' is discrimination' and 'b' parameters are difficulty parameter per each item.

Discussion.
Overall. Appropriate content is included, though additional implications and further research could be considered.
1. Per corrections noted above regarding parameter misidentification (discrimination, difficulty), correct all errors, e.g., Page 8. line 287 ". . . "Very true and often true" had higher discriminatory power."  This is incorrect.  These response options have higher scale values, i.e., higher 'difficulty'.
2. As noted above, consider the measurement concept of 'measurement invariance' when discussing the issue regarding the Danish scaling, and also, the International Test Commission Guidelines.
3. Consider an implications section or paragraph discussing the benefits for researchers and not only practitioners. This is an opportunity to point out the importance for scaling sensitivity, when IRT really makes a difference.
4. Future research could include measurement invariance studies such that Danish and Polish samples are combined and scaling is completed and item invariance models are tested.  This is critically important given the emphasis you are placing on cross-cultural differences.

Author Response

The Response to the Reviewer #2.

Dear Reviewer 2,

Thank you for all insightful suggestions to the previous version of the manuscript. Below, we showed how we addressed your suggestions. Where necessary, we included line numbers from the clean version of the manuscript. These changes are also highlighted in the second uploaded file with changes tracked.

We hope that the revised version of the manuscript addresses your reservations to the previous version of the manuscript.

Thank you for all comments and efforts to improve this submission,

The Authors

Reviewer #2: Abstract.
The Abstract provides an excellent summary of your research manuscript.  A few writing suggestions include the following.
1. replace ". . . assessed their personality 'impairments'" with ". . . assessed their personality 'functioning'"

The Authors: we have corrected the sentence according to the Reviewer’s suggestion.

Reviewer #2: 2. avoid writing in the first person, e.g., "We developed . . ."

The Authors: We rewrote the sentence to avoid writing in first person.

Reviewer #2: 3. ". . .implications for healthcare are being discussed."  ". . .implications for healthcare practice and research are discussed."

The Authors: We have corrected the sentence according to the Reviewer’s suggestion.

Reviewer #2: Keywords.
1. Consider keywords specific to the manuscript as well as keywords provided for the Weekers et al. (2023) publication; e.g., Polish population,
2. Item Response Theory, validity are irrelevant keywords (note my comment regarding the title.

The Authors: We have omitted phrases: “Item Response Theory” and “validity” in the keywords. We added “Polish population” to the keywords.

Reviewer #2: Title.
1. item response theory is merely the most appropriate modeling framework for completing the analyses, and does not merit emphasis in the title unless the method is somehow more than 'the method'. For this research publication, in my opinion mention of Item Response Theory in the title is somewhat distracting.

The Authors: Thank you for this suggestion. We have omitted the Item Response Theory in the revised title.

Reviewer #2: Introduction.
Overall. The introduction includes all substantive content. A few writing corrections are recommended.  Note a complex suggestion in item 3 below suggesting re-organization of key content.

  1. writing. page 2, line 72. ". . . LPFS-BF 2.0 should the unidimensional as intended . . ."  (should be unidimensional)

The Authors: Thank you. We corrected this typo in the revised version of the manuscript.

Reviewer #2: 2. page 2, line 86. Consider subheading paragraph 4 'Item Response Scaling'.  This is a central idea to the manuscript you want to provide as you emphasize the scaling technology.

The Authors: We added a subhead “Item Response Scaling”. Within this subsection we described benefits of IRT in developing severity benchmarks. The paragraph could be found in lines 104-131:

1.1.         Item response scaling

Normative values of the LPFS-BF 2.0 were frequently constructed based on the classical test theory (CTT) approaches which assume a linear relation between latent variable and test score, equal reliability across scale scores, and normal distribution as a basis of normalization procedures [24]. Moreover, the focus of CTT is on the test score more than on the item response scaling, and parameters such as reliability, dis-crimination, and location strongly rely on the sample [24-25]. An alternative approach to the CTT is Item Response Theory (IRT) which models the association between a sub-ject’s score on a latent variable and the probability of a particular response to an item [26]. IRT assumes that the person’s response to an item is a function of their location on a latent variable and of certain characteristics of the respective item [27]. Using in-formation functions for the item and the test score, IRT describes how measurement precision can vary across different levels of the latent variable [28]. It can estimate item properties independently of the sample used [28; for more details on the differ-ences between CTT and IRT see e.g., 29]. In general, IRT offers more precision in meas-urement of the person’s latent score and provides more insight into the properties of the items used in the measurement, which are estimated in a not-sample dependent way.

Cut-off scores or severity benchmarks are useful in supporting diagnosis and tracking change during treatment [30]. There are many methods to establish mean-ingful thresholds (e.g., a receiver curve operator [ROC], predictive modeling; see 30-31). IRT offers a method of determining the cut-off point based on the standard de-viation from the latent trait mean [10]. It offers more precision and less dependence on the sample in determining cut-off scores for clinical use. IRT could also better clarify the latent trait range measured by the items and the informative function of response categories used to answer an item [32]. Previous studies showed that IRT-based cut-offs outperform other methods (e.g, ROC or predictive modeling; 30), especially in cases when the prevalence of the phenomenon tested (here: severe personality im-pairment) is less than .30 [33] which is the case for personality disorders in the general population [2].

Reviewer #2: 3. Consider re-organizing the content of paragraphs 4 (page 2, line 86) and paragraph 5 (page 3, line 113).  The content of one paragraph is 'Item Response Scaling' (paragraph 3). The content of the next paragraph 4 is 'Lack of Cultural Invariance' in which you explain the need for a Polish scaling given that the Danish scaling is inappropriate, i.e., scaling is not invariant.  

The Authors: We added the paragraph in line 132-163:

1.2. Cultural invariance of the LPFS-BF 2.0

A recent study on a representative group of the Danish population used IRT to select severity benchmarks for the LPFS-BF 2.0 based on standard deviations from the mean of the latent variable of the LPFS-BF 2.0 total score [11]. It was the first study to suggest cut-off criteria for the LPFS-BF 2.0. Based on the distance from the mean value of the latent variable (θ) representing severity of personality psychopathology, the suggested score of 26 in the LPFS-BF 2.0 represented cut-off for mild level of personality impairments (1.0 standard deviation [SD] above the mean θ, T-score = 60), a score of 31 reflected a moderate level of personality pathology (1.5 SD above the latent mean; T-score = 65). Likewise, a score of 36 corresponded to a severe level of personality disfunction (2.0 SD above the latent mean; T-score = 70), whereas a score of 41 reflected an extreme level of personality pathology (2.5 SD above the latent mean; T-score = 75). These cut-offs were suggested to be appropriate for Denmark and other Nordic countries, and possibly also for Western countries [11]. However, previous studies had shown that the mean score of the LPFS-BF 2.0 for the Danish population was significantly lower compared to other European, American, and Asian countries [34]. This could indicate that severity benchmarks for the LPFS-BF 2.0 based on the Danish population could result in false positive detection of personality impairment when used in a different population. Although, Natoli et al. [34] showed the measurement invariance of the LPFS-BF 2.0 across tested counties (US, Italy, Denmark, United Arab Emirates], there is not stablished measurement invariance between the Polish and Danish versions of the LPFS-BF 2.0. Moreover, previous studies on the measurement invariance of the LPFS-BF 2.0 used confirmatory factor analysis procedure to establish measurement invariance [34-35]. Meanwhile, alternative IRT-based procedures exist and are suggested by the International Test Commission (ITC) which could better test for item invariance between populations [36]. Unfortunately, to our knowledge there is no study testing the measurement invariance of the IRT model of the LPFS-BF 2.0, which could be important for using IRT-derived cut-off between populations. In general, population-specific cut-offs, unless they are cross-validated across different regions, might lead to incorrect diagnoses or overdiagnosis, such as false positives, in countries with distinct cultural or social contexts. Thus, further studies verifying the suggested generalizability of the developed cut-offs are needed to improve the implementation of the LPFS-BF 2.0 in public healthcare service in other countries.

Reviewer #2: 4. Note, pertaining to lack of cultural invariance you may consider reference to the International Test Commission (ITC) https://www.intestcom.org/ guidelines and concepts/technologies, including IRT pertaining to scaling measures across populations from different cultures/languages.

The Authors: We referred to the ITC in lines 154-156:

Meanwhile, alternative IRT-based procedures exist and are suggested by the International Test Commission (ITC) which could better test for item invariance between populations [36].

Reviewer #2: Materials and Methods.
Overall. The methods are appropriate, though a few suggestions are provided.
1. Table 1.  The column titles need to be reversed. The population is the first column, the 'present study' is the second column.

The Authors: We reversed the names of the columns.

Reviewer #2: 2. Measures. For the PiCD, report the reliability coefficients and not only the references for those psychometrics.

The Authors: In the revised version we referred to the reliability of the PiCD in Polish validation by Cieciuch et al., 2022. Moreover, reliability for all PiCD subscales in the present study were added in l. 226-231:

Previous studies in the Polish population demonstrated factorial validity and high reliability (Cronbach’s α between .77 for Anankastia and .87 for Negative affect subscales) of the PiCD [42]. In the present study internal consistency for the measured pathological traits were: Negative affect: Cronbach’s α = .90, Disinhibition: Cronbach’s α = .91, Detachment: Cronbach’s α = .88, Dissociality: Cronbach’s α = .89, and Anankastia: Cronbach’s α = .80.

Reference:

Cieciuch, J.; Lakuta, P.; Strus, W.; Oltmanns, J.R.; Widiger, T. Assessment of personality disorder in the ICD-11 diagnostic system: Polish validation of the Personality Inventory for ICD-11. Psychiatr Pol 2022, 56(6), 1185-1202.

Reviewer #2: 3. page 5, paragraph 2. Your description of IRT modeling and model choice (graded response) needs revision. First, consider a more complete explanation of why the 'graded response model'.  Rather than merely ". . .because of the polytomous response scale", include the model constraint that response options vary on scale with respect to location only, i.e., they are equally sensitive.  

The Authors: We added more in-depth explanation for using GRM and use Vuong’s test to compare the usefulness of GRM with GPCM. The added sentences are in lines 257-266:

Item response theory parameters were estimated using the graded response model (GRM) because of the polytomous response scale [47]. GRM allows estimation of a discrimination parameter for each item and assumes that response options vary on the scale with respect to location only. In the GRM, item difficulties are calculated cumulatively by modeling the probability that an individual will respond to a given response category or higher. Other competing IRT models for polytomous responses include the generalized partial credit model which models the probability of responding to a specific response category directly (GPCM; 48). We evaluated whether GRM and GPCM were distinguishable for the LPFS-BF 2.0, and which model had better fit using Vuong test [49-50]. Based on the results of the Vuong test, we decided which model we would use in the analysis.

And in Result section (lines 304-311):

Based on the unidimensional structure of the LPFS-BF 2.0, we ran the graded response IRT model which resulted in a good fit to the data (RMSEA = .07; 95% CI for RMSEA = [.06 – .08]; SRMR = .05; TLI = .98; CFI = .98). An alternative approach, namely GPCM, also had good fit (RMSEA = .07; 95% CI for RMSEA = [.06 – .08]; SRMR = .05; TLI = .98; CFI = .98). However, the Vuong test indicated that these two models were distinguishable (w2 = .147; p < .001) and that the GRM model had better fit compared to the GPCM (z = 6.59; p < .001). Thus, we proceeded with the GRM IRT model of the LPFS-BF 2.0.   

Reviewer #2: 4. Preceding note 2 above points out a serious error in your writing. Page 5, lines 189-190. You state, "Item characteristic curve was used to visualize the discriminatory properties of the response scale for each item."  NOTE.  Each item has a different 'difficulty' parameter, not discrimination parameter, discrimination is constant across items within the graded response model.  

The Authors: Thank you for these important suggestions. We corrected the sentences as below:

  1. 267-270: Then, we evaluated the discrimination parameter for each item (a1), as well as the item difficulty parameters (b1 – b3) indicating for a particular category k of the response scale how an item reflects the level of the attribute at which patients have a 50% likelihood of scoring a category lower than k versus category k or higher [26].

Reviewer #2: Results.

Overall. The results are complete, but need some revision, particularly correction to Table 2 title.
1. Refer to comment 4 above regarding graded response model parameters.  Table 2 title has parameter identification reversed, 'a1' is discrimination' and 'b' parameters are difficulty parameter per each item.

The Authors: The title of Table 2 was corrected according to the Reviewer’s suggestion.

Reviewer #2: Discussion.

Overall. Appropriate content is included, though additional implications and further research could be considered.
1. Per corrections noted above regarding parameter misidentification (discrimination, difficulty), correct all errors, e.g., Page 8. line 287 ". . . "Very true and often true" had higher discriminatory power."  This is incorrect.  These response options have higher scale values, i.e., higher 'difficulty'.

The Authors: We have corrected all errors in differentiations between discrimination and difficulty parameters (lines 383-386): However, we showed that the response options: “Sometimes true or somewhat true” and “Very true and often true” had higher difficulty. It was indicated by the high difficulty parameters (b2-b3) for the LPFS-BF 2.0 items. This was in line with the previous observations by Weekers et al. [11].

Reviewer #2: 2. As noted above, consider the measurement concept of 'measurement invariance' when discussing the issue regarding the Danish scaling, and also, the International Test Commission Guidelines.

The Authors: We referred to measurement invariance in Discussion in lines 409-416:

Natoli et al. [34] showed that Danish participants reported the lowest latent mean of the LPFS-BF 2.0 across seven countries (Canada, Chile, Denmark, Germany, Italy, United States of America, and United Arab Emirates). Thus, we believe that the generalizability of the LPFS-BF 2.0 cut-offs developed using the Danish sample may be in fact too sensitive and increase the rate of false positive diagnoses of personality impairment in other populations. Particularly, using cut-off score derived from Danish population may be not proper because of lack of established measurement invariance between the Polish and Danish version of the LPFS-BF 2.0.

And in lines 478-494:

In the present study, we developed severity benchmarks for application of the LPFS-BF 2.0 in clinical practice in Poland. We based our cut-off scores on a sample of Poles representing the general population in terms of the proportions of sex and age and on the IRT-based distances from the latent mean score. The proposed severity benchmarks were more conservative in screening for personality impairments compared to the cut-offs developed by Weekers et al. [11], possibly omitting false positive categorizations. The differences in the suggested cut-off points, particularly for mild impairment could result from lack of cultural invariance of the LPFS-BF 2.0 between the Polish and Danish populations. Possible sources of such differences could be sought in the differences in responding styles, differences in social desirability or social norms underlying the responses, or different reference points when making statement about oneself in the two populations [60]. When applied directly to the Polish population, Weekers et al. [11] severity benchmarks could indicate that individuals without personality impairments would be classified as mildly or moderately impaired in the Polish context. Thus, deriving the cut-off points based on the Polish sample could make it possible to overcome the errors connected with applying severity benchmarks from another culture without testing measurement invariance as a precondition of such application.

Reviewer #2: 3. Consider an implications section or paragraph discussing the benefits for researchers and not only practitioners. This is an opportunity to point out the importance for scaling sensitivity, when IRT really makes a difference.

The Authors: We have added the section: 4.3. Implications for research (ll. 565-580):

The present study supports the usefulness of IRT to derive severity benchmarks for clinical measures [32]. IRT analysis of the LPFS-BF 2.0 items could, moreover, indicate how precise are all the items in mapping the range of the latent trait of personality impairment and how to interpret the test taker’s responses. For example, the present study showed that test scores of the LPFS-BF 2.0 reflect theta range from -2 to 3, whereas in Weekers et al. [11], the theta range was narrower (0 to 3). This observation could have consequences for understanding the latent variable measured with the LPFS-BF 2.0. The present study suggests that low scores of the LPFS-BF 2.0 may represent the opposite pole (healthy personality functioning) with regard to severe personality impairment linked with high scores on the LPFS-BF 2.0. However, in other populations, the LPFS-BF 2.0 seems to map the lack of or intensity of personality impairment only. Another important goal of future studies is to establish measurement in-variance for the LPFS-BF 2.0 using IRT analysis [36]. Moreover, the IRT-derived model of measuring personality impairment using the LPFS-BF 2.0 makes it possible to over-come the limitations of conventional, CTT-based application of the LPFS-BF 2.0 (e.g., sample dependence).

Reviewer #2: 4. Future research could include measurement invariance studies such that Danish and Polish samples are combined and scaling is completed and item invariance models are tested.  This is critically important given the emphasis you are placing on cross-cultural differences.

The Authors: We added the following comment to the limitation section:

  • 528-530: Lastly, future studies should test measurement invariance of the LPFS-BF 2.0 in Polish and Danish population to better explain the reasons for the differences in the cut-off scores obtained (e.g., the role of cultural context).

Reviewer 3 Report

Comments and Suggestions for Authors

Dear Authors,

Thank you for providing me with the opportunity to read this interesting piece of paper. Below, I have listed my comments:

  • The introduction references both the DSM-5 and ICD-11, yet the differences between the two frameworks aren't fully addressed. A short explanation clarifying that the DSM-5 emphasizes personality traits while the ICD-11 takes a broader approach to assessing personality could benefit readers who are less familiar with these systems. This would also help clarify the importance of both models within the context of the study.

    The discussion on the development of cut-off criteria for the LPFS-BF 2.0 in Denmark could be expanded by addressing the limitations of these benchmarks. For instance, it could emphasize that population-specific cut-offs, if not cross-validated across different regions, might lead to incorrect diagnoses or overdiagnosis, such as false positives, in countries with distinct cultural or social contexts.

    The introduction briefly mentions the Polish version of the LPFS-BF 2.0, but it could benefit from additional context about why this specific population was chosen. Are there particular cultural or healthcare-related factors in Poland that might influence how this scale is applied? Providing a brief explanation for selecting this population would better engage readers with the study’s rationale.

    The introduction quickly transitions from personality disorders to the LPFS-BF 2.0 and the IRT approach. Including a sentence or two to smoothly link these sections would enhance the flow and provide a clearer connection between these key concepts.

    In the methods section, it would be helpful to briefly clarify whether relationship status was assessed at the time of the survey or over a lifetime. This distinction could provide important context, if applicable.

    Including any relevant exclusion criteria for participants could add clarity to the methodology. For example, it would be useful to mention whether participants with missing data or specific mental health conditions were excluded from the study.

    When discussing the LPFS-BF 2.0 and PiCD, it would be beneficial to mention whether these tools have previously been translated and validated in Polish, as this would provide context for their current use.

    The comparison between the benchmarks in this study and those by Weekers et al. is crucial, but a bit more clarification could help illustrate why these differences have clinical significance. For example, a more detailed explanation of how a higher threshold (as used in this study) may reduce false positives and aid in clinical decision-making would be valuable.

    While the study underscores the potential value of these benchmarks for screening, it would be beneficial to explore how they might be practically applied in clinical settings. For example, how could these severity benchmarks be incorporated into current screening protocols, and what challenges might clinicians encounter when using the LPFS-BF 2.0 with the newly proposed cut-offs?

    The implications section provides a solid overview, but expanding on the potential impact of these findings on healthcare systems would further emphasize the significance of these benchmarks. For instance, what types of interventions could be improved or tailored by incorporating these severity benchmarks into clinical practice?

    In some areas, the terms "subclinical personality impairment" and "mild personality impairment" are used interchangeably. Using consistent terminology throughout the discussion would help improve clarity and make the text easier to follow.

  • I hope this feedback is helpful.

Author Response

The Response to the Reviewer #3

Dear Reviewer 3,

Thank you for all insightful suggestions to the previous version of the manuscript. Below, we showed how we addressed your suggestions. Where necessary, we included line numbers from the clean version of the manuscript. These changes are also highlighted in the second uploaded file with changes tracked.

We hope that the revised version of the manuscript addresses your reservations to the previous version of the manuscript.

Thank you for all comments and efforts to improve this submission,

The Authors

Reviewer #3: The introduction references both the DSM-5 and ICD-11, yet the differences between the two frameworks aren't fully addressed. A short explanation clarifying that the DSM-5 emphasizes personality traits while the ICD-11 takes a broader approach to assessing personality could benefit readers who are less familiar with these systems. This would also help clarify the importance of both models within the context of the study.

The Authors: Thank you for these suggestions. We provide more information about the DSM-5 and ICD-11 conceptualizations of personality pathology (regarding similarities and differences) in lines 38-53:

Because of shortcomings of the previous categorial approach to diagnose personality disorders, the recent editions of the most important psychiatric manuals (Diagnostic and Statistical Manual of Mental Disorders [DSM-5], and the World Health Organization [WHO] International Classification of Diseases [ICD-11]) proposed trait-based personality pathology models [4]. The Alternative Model for Personality Disorders (AMPD; [5]) introduced in DSM-5 recognized two criteria of personality disorder: (a) impairments in self and interpersonal functioning (criterion A), and (b) the presence of one or more pathological traits (negative affect, detachment, disinhibition, antagonism, psychoticism (criterion B). The ICD-11 model of pathological personality traits includes an assessment of personality psychopathology severity (mild, moderate, and severe) and trait domain specifiers including negative affect, detachment, disinhibition, dissociality (an equivalent of antagonism in DSM-5), and anankastia, but not psychoticism [6]. Despite some differences in conceptualization of the severity of personality impairment (criterion A) and the role of pathological trait domains in personality disorder diagnosis (criterion B), significant similarity has been shown between trait-based models of personality in ICD-11 and DSM-5 [4, 7-8].

Reviewer #3: The discussion on the development of cut-off criteria for the LPFS-BF 2.0 in Denmark could be expanded by addressing the limitations of these benchmarks. For instance, it could emphasize that population-specific cut-offs, if not cross-validated across different regions, might lead to incorrect diagnoses or overdiagnosis, such as false positives, in countries with distinct cultural or social contexts.

The Authors: The discussion of the limitation of cut-off scores developed for Danish population were expanded in lines 133-163:

A recent study on a representative group of the Danish population used IRT to select severity benchmarks for the LPFS-BF 2.0 based on standard deviations from the mean of the latent variable of the LPFS-BF 2.0 total score [11]. It was the first study to suggest cut-off criteria for the LPFS-BF 2.0. Based on the distance from the mean value of the latent variable (θ) representing severity of personality psychopathology, the suggested score of 26 in the LPFS-BF 2.0 represented cut-off for mild level of personality impairments (1.0 standard deviation [SD] above the mean θ, T-score = 60), a score of 31 reflected a moderate level of personality pathology (1.5 SD above the latent mean; T-score = 65). Likewise, a score of 36 corresponded to a severe level of personality disfunction (2.0 SD above the latent mean; T-score = 70), whereas a score of 41 reflected an extreme level of personality pathology (2.5 SD above the latent mean; T-score = 75). These cut-offs were suggested to be appropriate for Denmark and other Nordic countries, and possibly also for Western countries [11]. However, previous studies had shown that the mean score of the LPFS-BF 2.0 for the Danish population was significantly lower compared to other European, American, and Asian countries [34]. This could indicate that severity benchmarks for the LPFS-BF 2.0 based on the Danish population could result in false positive detection of personality impairment when used in a different population. Although, Natoli et al. [34] showed the measurement invariance of the LPFS-BF 2.0 across tested counties (US, Italy, Denmark, United Arab Emirates], there is not stablished measurement invariance between the Polish and Danish versions of the LPFS-BF 2.0. Moreover, previous studies on the measurement invariance of the LPFS-BF 2.0 used confirmatory factor analysis procedure to establish measurement invariance [34-35]. Meanwhile, alternative IRT-based procedures exist and are suggested by the International Test Commission (ITC) which could better test for item invariance between populations [36]. Unfortunately, to our knowledge there is no study testing the measurement invariance of the IRT model of the LPFS-BF 2.0, which could be important for using IRT-derived cut-off between populations. In general, population-specific cut-offs, unless they are cross-validated across different regions, might lead to incorrect diagnoses or overdiagnosis, such as false positives, in countries with distinct cultural or social contexts. Thus, further studies verifying the suggested generalizability of the developed cut-offs are needed to improve the implementation of the LPFS-BF 2.0 in public healthcare service in other countries.

Reviewer #3: The introduction briefly mentions the Polish version of the LPFS-BF 2.0, but it could benefit from additional context about why this specific population was chosen. Are there particular cultural or healthcare-related factors in Poland that might influence how this scale is applied? Providing a brief explanation for selecting this population would better engage readers with the study’s rationale.

The Authors: The Polish context was added in lines 174-181:

The Polish population was selected because of the initial stage of the research on the validity of the LPFS-BF 2.0 in the Polish population and because of the difficulties in personality impairment and disorder diagnosis in Polish psychiatric healthcare [37]. Polish psychiatric care units tend to focus on the disorders from the I axis and lack valid measures which could foster diagnosis of comorbid personality impairments [22]. Moreover, comprehensive validation of the LPFS-BF 2.0 could warrant its proper use in basic psychiatric screening as recommended by international councils [16].

Reviewer #3: The introduction quickly transitions from personality disorders to the LPFS-BF 2.0 and the IRT approach. Including a sentence or two to smoothly link these sections would enhance the flow and provide a clearer connection between these key concepts.

The Authors: The introduction was re-organized. We added subheading Item response scaling, and Cultural invariance of the LPFS-BF 2.0. We hope that these changes make the transitions from personality disorder to IRT approach easier to follow.

Reviewer #3: In the methods section, it would be helpful to briefly clarify whether relationship status was assessed at the time of the survey or over a lifetime. This distinction could provide important context, if applicable.

The Authors: The relationship status was measures at the moment of the study (l. 198-200). We did not measure a lifetime number of relationships. In the limitation section we added one sentence addressing this issue:

  1. 526-528:

Moreover, the participants could report on lifetime romantic involvement to precisely establish the association between personality impairment and problems with entering into romantic relationships.

Reviewer #3: Including any relevant exclusion criteria for participants could add clarity to the methodology. For example, it would be useful to mention whether participants with missing data or specific mental health conditions were excluded from the study.

The Authors: In the revised version we added the following sentences addressing this issue:

  1. 154-155: The only exclusion criterion was age below 18. No particular exclusion criteria regarding mental health were applied.
  2. 236-238: The participants were informed that they should answer all questions of the on-line form, resulting in no missing data in the present study.

Reviewer #3: When discussing the LPFS-BF 2.0 and PiCD, it would be beneficial to mention whether these tools have previously been translated and validated in Polish, as this would provide context for their current use.

The Authors: Both the LPFS-BF 2.0 and PiCD were adapted in Polish. In the Measures section we included references to Polish validation studies.

  1. Lakuta, P.; Cieciuch, J.; Strus, W.; Hutsebaut, J. Level of Personality Functioning Scale-Brief Form 2.0: Validity and reliability of the Polish adaptation. Psychiatr Pol 2022, 270,1–14. https://doi.org/ 10.12740/PP/OnlineFirst/145912
  2. Cieciuch, J.; Lakuta, P.; Strus, W.; Oltmanns, J.R.; Widiger, T. Assessment of personality disorder in the ICD-11 diagnostic system: Polish validation of the Personality Inventory for ICD-11. Psychiatr Pol 2022, 56(6), 1185-1202.

Reviewer #3: The comparison between the benchmarks in this study and those by Weekers et al. is crucial, but a bit more clarification could help illustrate why these differences have clinical significance. For example, a more detailed explanation of how a higher threshold (as used in this study) may reduce false positives and aid in clinical decision-making would be valuable.

The Authors: We devoted one paragraph to discuss these problems.

Lines 397-416:

First, the cut-offs developed in the present study could be more conservative with regard to the screening function of the LPFS-BF 2.0. However, a closer inspection of the cut-offs developed by Weekers et al. [11] indicated crossing the threshold between “healthy” personality and mild personality impairment required only two items out of twelve scored 3 (“Sometimes true or somehow true”) and ten items scored 2 (“Some-times false or somehow false”). In our case, a mild level of impairment required at least eight items scored 3 and the remaining four scored 4 to be screened as being at risk of mild personality impairment. In comparison, cut-offs developed using the Dan-ish sample may be too sensitive (e.g., resulting in false positives) when applied to a different cultural context (e.g., Polish), whereas those developed in our sample may be more conservative in screening for personality impairment in Poland. Second, the dif-ferences could result from nation-level differences in personality problems between the Polish and Danish populations. Natoli et al. [34] showed that Danish participants re-ported the lowest latent mean of the LPFS-BF 2.0 across seven countries (Canada, Chile, Denmark, Germany, Italy, United States of America, and United Arab Emirates). Thus, we believe that the generalizability of the LPFS-BF 2.0 cut-offs developed using the Danish sample may be in fact too sensitive and increase the rate of false positive diagnoses of personality impairment in other populations. Particularly, using cut-off score derived from Danish population may be not proper because of lack of established measurement invariance between the Polish and Danish version of the LPFS-BF 2.0.

And also mentioned int in Limitation section:

Lines 481-493:

The proposed severity benchmarks were more conservative in screening for personality impairments compared to the cut-offs developed by Weekers et al. [11], possibly omitting false positive categorizations. The differences in the suggested cut-off points, particularly for mild impairment could result from lack of cultural invariance of the LPFS-BF 2.0 between the Polish and Danish populations. Possible sources of such differences could be sought in the differences in responding styles, differences in social desirability or social norms underlying the responses, or different reference points when making statement about oneself in the two populations [60]. When applied directly to the Polish population, Weekers et al. [11] severity benchmarks could indicate that individuals without personality impairments would be classified as mildly or moderately impaired in the Polish context. Thus, deriving the cut-off points based on the Polish sample could make it possible to overcome the errors connected with applying severity benchmarks from another culture without testing measurement invariance as a precondition of such application.

Reviewer #3: While the study underscores the potential value of these benchmarks for screening, it would be beneficial to explore how they might be practically applied in clinical settings. For example, how could these severity benchmarks be incorporated into current screening protocols, and what challenges might clinicians encounter when using the LPFS-BF 2.0 with the newly proposed cut-offs?

The Authors: We addressed these issues in lines 544-548:

The LPFS-BF 2.0 could be used to improve psychotherapeutic diagnosis and during the psychiatric diagnosis process. For example, the LPFS-BF 2.0 could be incorporated in the model of diagnosis of personality disorders or in the general standards of psychiatric diagnosis to improve the screening for personality psychopathology which is neglected in Polish healthcare [37].

And 555-560:

However, using the LPFS-BF 2.0 as an additional screening tool may result in some difficulties. For example, they could result in more comorbidity in diagnosis. This could require the development of new recommendations on how to prepare a treatment protocol in case of comorbid personality impairment. In general, such changes in healthcare organization could result in a more detailed and individualized approach to the patient’s psychopathology and in the development of more effective care.

Reviewer #3: The implications section provides a solid overview, but expanding on the potential impact of these findings on healthcare systems would further emphasize the significance of these benchmarks. For instance, what types of interventions could be improved or tailored by incorporating these severity benchmarks into clinical practice?

The Authors: We added the following sentences to address this issue:

Line 552-555: Based on screening for personality impairment, clinicians or psychotherapists could include additional interventions or techniques which could address their patient’s personality impairments (pharmacological interventions, psychodynamic or schema therapy, assistance of psychiatric nurses) [37].

Reviewer #3: In some areas, the terms "subclinical personality impairment" and "mild personality impairment" are used interchangeably. Using consistent terminology throughout the discussion would help improve clarity and make the text easier to follow.

The Authors: We used “mild/subclinical” term to be consistent with previous study on the LPFS-BF 2.0 severity benchmarks (Weekers et al., 2023). However, according to the Reviewer’s suggestion, we focus on term “mild” because we applied no clinical criterion for distinguishing individuals with clinical level of severity of the personality impairment as measured with the LPFS-BF 2.0.

Reviewer 4 Report

Comments and Suggestions for Authors

Severity benchmarks for the Level of Personality Functioning Scale – Brief Form 2.0 (LPFS-BF 2.0) in Polish adults: Item Response Theory Perspective

Introduction

1. The first paragraph is difficult for readers who are not in the field as it introduces several models and guidelines, such as AMPD and DSM-5. So, it is necessary to consider whether this model needs to be introduced. If it does, a softer approach may be needed to introduce these terms.

2. Writers tend to use long sentences to describe their ideas. However, these long sentences can be confusing for the reader.

3. As with any article related to IRT, it is appropriate to state the advantages of measurement using IRT over CTT, especially in this study's construct [see page 2, L89-91].

4. The author should briefly discuss the advantages of calculating cutoff scores based on latent mean. This is because critics insist that such a procedure assumes that the score distribution is normal, even though, in reality, the score distribution is usually skewed or multimodal [page 3, second paragraph]

Materials and Method

1. Revisit Table 1 - statistics of the population were small. 2. In general, this section is well-written.

Results

1. The TLI = 1.00 and CFI = 1.00 can indicate overfitting of the model - it measures noise?

2. It might be helpful if the conversion from the theta scale to the raw score scale was shown to facilitate understanding. More explanations on how to obtain important values such as SD might also help readers understand more deeply.

Discussion

1. There is a significant difference between the cutoff scores established in this study and those by Weekers et al. Given that the characteristics of patients in each category should be similar, regardless of where they are located, it may be worth discussing the extent of the impact of these findings.

2. What is meant by a cutoff score that is 'too sensitive'? The common understanding is that when the cutoff scores are established, they are close. This was not seen in the study by Weekers et al.

3. However, we showed that the response options "Sometimes true or somewhat true" and "Very true and often true" had higher discriminatory power. It might be good to review the evidence from the study findings that led to this conclusion [Page 8, L286-287]. State again the statistics that led to this conclusion and its implications.

4. The authors wrote, "We also noticed that the cutoffs developed for the LPFS-BF 2.0 were positively associated with individuals' actual intentions to undertake psychotherapy" [Page 9, L342-343]. It might be good to review the evidence from the study findings that led to this conclusion.

5. In studies related to the setting of cutoff scores, there is a strong opinion that emphasizes that the establishment of cutoff scores should also involve discussing the study results with experts to justify the resulting cutoff scores. To what extent does the author feel the procedure comprehensively determines patient classification?

Overall

1. References: This article includes some recent and relevant studies but is perhaps too dependent on Weekers et al. (2023). Perhaps including other international studies would provide richer descriptions/explanations/justifications.

2. Research Design: As the authors mention, the exclusion of the clinical population might restrict the use of these cutoff scores.

3. Presentation of Results: It is quite difficult for me to understand some parts of the manuscript, especially when involving very specific terms, whether in the medical or psychometric fields, that may be unfamiliar to the reader

Author Response

The Response to the Reviewer #4

Dear Reviewer 4,

Thank you for all insightful suggestions to the previous version of the manuscript. Below, we showed how we addressed your suggestions. Where necessary, we included line numbers from the clean version of the manuscript. These changes are also highlighted in the second uploaded file with changes tracked.

We hope that the revised version of the manuscript addresses your reservations to the previous version of the manuscript.

Thank you for all comments and efforts to improve this submission,

The Authors

Reviewer #4: Introduction

  1. The first paragraph is difficult for readers who are not in the field as it introduces several models and guidelines, such as AMPD and DSM-5. So, it is necessary to consider whether this model needs to be introduced. If it does, a softer approach may be needed to introduce these terms.

The Authors: We introduce more information about the models of personality disorder diagnosis in psychiatric manuals. They were added in lines 38-57:

Because of shortcomings of the previous categorial approach to diagnose personality disorders, the recent editions of the most important psychiatric manuals (Diagnostic and Statistical Manual of Mental Disorders [DSM-5], and the World Health Organization [WHO] International Classification of Diseases [ICD-11]) proposed trait-based personality pathology models [4]. The Alternative Model for Personality Disorders (AMPD; [5]) introduced in DSM-5 recognized two criteria of personality disorder: (a) impairments in self and interpersonal functioning (criterion A), and (b) the presence of one or more pathological traits (negative affect, detachment, disinhibition, antagonism, psychoticism (criterion B). The ICD-11 model of pathological personality traits includes an assessment of personality psychopathology severity (mild, moderate, and severe) and trait domain specifiers including negative affect, detachment, disinhibition, dissociality (an equivalent of antagonism in DSM-5), and anankastia, but not psychoticism [6]. Despite some differences in conceptualization of the severity of personality impairment (criterion A) and the role of pathological trait domains in personality disorder diagnosis (criterion B), significant similarity has been shown between trait-based models of personality in ICD-11 and DSM-5 [4, 7-8]. A significant difference from the previous categorical approach with arbitrary thresholds for clinically relevant personality disorder pertains to the focus on a dimensional approach to personality functioning. This creates a need for proper continuous assessment of personality functioning making it possible to differentiate impaired from healthy personality functioning [4].

Reviewer #4: 2. Writers tend to use long sentences to describe their ideas. However, these long sentences can be confusing for the reader.

The Authors: The revised manuscript was checked and corrected by the professional English editor to make the sentences less confusing.

Reviewer #4: 3. As with any article related to IRT, it is appropriate to state the advantages of measurement using IRT over CTT, especially in this study's construct [see page 2, L89-91].

The Authors: In the revised version of the manuscript, we devoted one section (1.1. Item Response Scaling) in which, we added broader description of the IRT and its advantages over CTT. The section is in lines 105-131:

Normative values of the LPFS-BF 2.0 were frequently constructed based on the classical test theory (CTT) approaches which assume a linear relation between latent variable and test score, equal reliability across scale scores, and normal distribution as a basis of normalization procedures [24]. Moreover, the focus of CTT is on the test score more than on the item response scaling, and parameters such as reliability, dis-crimination, and location strongly rely on the sample [24-25]. An alternative approach to the CTT is Item Response Theory (IRT) which models the association between a sub-ject’s score on a latent variable and the probability of a particular response to an item [26]. IRT assumes that the person’s response to an item is a function of their location on a latent variable and of certain characteristics of the respective item [27]. Using in-formation functions for the item and the test score, IRT describes how measurement precision can vary across different levels of the latent variable [28]. It can estimate item properties independently of the sample used [28; for more details on the differ-ences between CTT and IRT see e.g., 29]. In general, IRT offers more precision in meas-urement of the person’s latent score and provides more insight into the properties of the items used in the measurement, which are estimated in a not-sample dependent way.

Cut-off scores or severity benchmarks are useful in supporting diagnosis and tracking change during treatment [30]. There are many methods to establish meaningful thresholds (e.g., a receiver curve operator [ROC], predictive modeling; see 30-31). IRT offers a method of determining the cut-off point based on the standard deviation from the latent trait mean [10]. It offers more precision and less dependence on the sample in determining cut-off scores for clinical use. IRT could also better clarify the latent trait range measured by the items and the informative function of response categories used to answer an item [32]. Previous studies showed that IRT-based cut-offs outperform other methods (e.g, ROC or predictive modeling; 30), especially in cases when the prevalence of the phenomenon tested (here: severe personality impairment) is less than .30 [33] which is the case for personality disorders in the general population [2].

Reviewer #4: 4. The The Author should briefly discuss the advantages of calculating cutoff scores based on latent mean. This is because critics insist that such a procedure assumes that the score distribution is normal, even though, in reality, the score distribution is usually skewed or multimodal [page 3, second paragraph]

 The Authors: The second part of the section 1.1. Item response scaling refers to this issue.

  1. 121-131: Cut-off scores or severity benchmarks are useful in supporting diagnosis and tracking change during treatment [30]. There are many methods to establish meaningful thresholds (e.g., a receiver curve operator [ROC], predictive modeling; see 30-31). IRT offers a method of determining the cut-off point based on the standard deviation from the latent trait mean [10]. It offers more precision and less dependence on the sample in determining cut-off scores for clinical use. IRT could also better clarify the latent trait range measured by the items and the informative function of response categories used to answer an item [32]. Previous studies showed that IRT-based cut-offs outperform other methods (e.g, ROC or predictive modeling; 30), especially in cases when the prevalence of the phenomenon tested (here: severe personality impairment) is less than .30 [33] which is the case for personality disorders in the general population [2].

Moreover, we added a section Implications for research in which we also show the advantages of IRT-based (latent variable-based) derivation of the cut-off points.

ll.566-580:

The present study supports the usefulness of IRT to derive severity benchmarks for clinical measures [32]. IRT analysis of the LPFS-BF 2.0 items could, moreover, indicate how precise are all the items in mapping the range of the latent trait of personality impairment and how to interpret the test taker’s responses. For example, the present study showed that test scores of the LPFS-BF 2.0 reflect theta range from -2 to 3, whereas in Weekers et al. [11], the theta range was narrower (0 to 3). This observation could have consequences for understanding the latent variable measured with the LPFS-BF 2.0. The present study suggests that low scores of the LPFS-BF 2.0 may repre-sent the opposite pole (healthy personality functioning) with regard to severe person-ality impairment linked with high scores on the LPFS-BF 2.0. However, in other popu-lations, the LPFS-BF 2.0 seems to map the lack of or intensity of personality impair-ment only. Another important goal of future studies is to establish measurement in-variance for the LPFS-BF 2.0 using IRT analysis [36]. Moreover, the IRT-derived model of measuring personality impairment using the LPFS-BF 2.0 makes it possible to over-come the limitations of conventional, CTT-based application of the LPFS-BF 2.0 (e.g., sample dependence).

Reviewer #4:

Materials and Method

  1. Revisit Table 1 - statistics of the population were small. 2. In general, this section is well-written.

The Authors: Table 1 was revised according to the Reviewer’s suggestion. 

Reviewer #4:

Results

  1. The TLI = 1.00 and CFI = 1.00 can indicate overfitting of the model - it measures noise?

The Authors: Thank you for this observation. The TLI, CFI, and RMSEA indices could be inflated because of the estimation method we have used. DWLS resulted in lower RMSEA and larger CF and TLI compared to other estimation method (e.g., ML; Xia & Jang, 2019). Second, our results from CFA were similar to those obtained in Danish sample (TLI = .99, and CFI = .99; Weekers et al., 2023). However, we also used other methods to test unidimensionality (parallel test and Velincer’s MAP test). Thus, based on these numerous approaches, we concluded that unidimensional model fitted the data properly.

Xia, Y., Yang, Y. RMSEA, CFI, and TLI in structural equation modeling with ordered categorical data: The story they tell depends on the estimation methods. Behav Res 51, 409–428 (2019). https://doi.org/10.3758/s13428-018-1055-2

Reviewer #4: 2. It might be helpful if the conversion from the theta scale to the raw score scale was shown to facilitate understanding. More explanations on how to obtain important values such as SD might also help readers understand more deeply.

The Authors: Thank you for this suggestion. Theta scores with corresponding expected LPFS-BF 2.0 scores are given in the Figure 1B. However, for the sake of clarity we also added Table A3 in the appendix which consists of the theta levels and corresponding expected LPFS-BF 2.0 scores.

Reviewer #4: Discussion

  1. There is a significant difference between the cutoff scores established in this study and those by Weekers et al. Given that the characteristics of patients in each category should be similar, regardless of where they are located, it may be worth discussing the extent of the impact of these findings.

The Authors: The reasons of the differences were discussed more in-depth in lines 478-494:

In the present study, we developed severity benchmarks for application of the LPFS-BF 2.0 in clinical practice in Poland. We based our cut-off scores on a sample of Poles representing the general population in terms of the proportions of sex and age and on the IRT-based distances from the latent mean score. The proposed severity benchmarks were more conservative in screening for personality impairments compared to the cut-offs developed by Weekers et al. [11], possibly omitting false positive categorizations. The differences in the suggested cut-off points, particularly for mild impairment could result from lack of cultural invariance of the LPFS-BF 2.0 between the Polish and Danish populations. Possible sources of such differences could be sought in the differences in responding styles, differences in social desirability or social norms underlying the responses, or different reference points when making statement about oneself in the two populations [60]. When applied directly to the Polish population, Weekers et al. [11] severity benchmarks could indicate that individuals without personality impairments would be classified as mildly or moderately impaired in the Polish context. Thus, deriving the cut-off points based on the Polish sample could make it possible to overcome the errors connected with applying severity benchmarks from another culture without testing measurement invariance as a precondition of such application.

And in lines 569-580:

For example, the present study showed that test scores of the LPFS-BF 2.0 reflect theta range from -2 to 3, whereas in Weekers et al. [11], the theta range was narrower (0 to 3). This observation could have consequences for understanding the latent variable measured with the LPFS-BF 2.0. The present study suggests that low scores of the LPFS-BF 2.0 may represent the opposite pole (healthy personality functioning) with regard to severe personality impairment linked with high scores on the LPFS-BF 2.0. However, in other populations, the LPFS-BF 2.0 seems to map the lack of or intensity of personality impairment only. Another important goal of future studies is to establish measurement invariance for the LPFS-BF 2.0 using IRT analysis [36]. Moreover, the IRT-derived model of measuring personality impairment using the LPFS-BF 2.0 makes it possible to overcome the limitations of conventional, CTT-based application of the LPFS-BF 2.0 (e.g., sample dependence).

Reviewer #4: 2. What is meant by a cutoff score that is 'too sensitive'? The common understanding is that when the cutoff scores are established, they are close. This was not seen in the study by Weekers et al.

The Authors: Thank you for this suggestion. We added th following explanation in the parenthesis:

  • 404-407: In comparison, cut-offs developed using the Danish sample may be too sensitive (e.g., resulting in false positives) when applied to a different cultural context (e.g., Polish), whereas those developed in our sample may be more conservative in screening for personality impairment in Poland

Reviewer #4: 3. However, we showed that the response options "Sometimes true or somewhat true" and "Very true and often true" had higher discriminatory power. It might be good to review the evidence from the study findings that led to this conclusion [Page 8, L286-287]. State again the statistics that led to this conclusion and its implications.

The Authors: We added the following sentence to clarify this issue:

  • 383-85: we showed that the response options: “Sometimes true or somewhat true” and “Very true and often true” had higher difficulty. It was indicated by the high difficulty parameters (b2-b3) for the LPFS-BF 2.0 items (see table 2)

Reviewer #4: 4. The Authors wrote, "We also noticed that the cutoffs developed for the LPFS-BF 2.0 were positively associated with individuals' actual intentions to undertake psychotherapy" [Page 9, L342-343]. It might be good to review the evidence from the study findings that led to this conclusion.

The Authors: We added the following sentences to clarify it:

  • 458-461: We also noticed that the cut-offs developed for the LPFS-BF 2.0 were positively associated with individuals’ actual intentions to undertake psychotherapy. Individuals classified as at risk of mild, moderate, or severe personality impairment reported higher frequency of psychotherapy seeking compared to those who were classified as “healthy”.

Reviewer #4: 5. In studies related to the setting of cutoff scores, there is a strong opinion that emphasizes that the establishment of cutoff scores should also involve discussing the study results with experts to justify the resulting cutoff scores. To what extent does the The Author feel the procedure comprehensively determines patient classification?

The Authors: We added the following suggestion regarding the Reviewer’s observation:

  • 549-551: However, the developed severity benchmarks should be previously consulted with experts in personality disorders diagnosis to further validate them. They should be cross-validated with clinician’s judgment based on in-depth diagnosis of personality pathology.

Reviewer #4: Overall

  1. References: This article includes some recent and relevant studies but is perhaps too dependent on Weekers et al. (2023). Perhaps including other international studies would provide richer descriptions/explanations/justifications.

The Authors: In the revised version we added about 30 new references. We rly strongly on Weekers et al (2023) because it was only study using IRT to establish cut-off scores for the LPFS-BF 2.0.

Reviewer #4: 2. Research Design: As the The Authors mention, the exclusion of the clinical population might restrict the use of these cutoff scores.

The Authors: We add this as a limitation:

  • 496-498: First, in the studied sample there were no cases of reported personality disorders. Thus, future studies should compare clinical and non-clinical samples examining whether individuals with diagnosed personality disorder have higher scores on the LPFS-BF 2.0.

Reviewer #4: 3. Presentation of Results: It is quite difficult for me to understand some parts of the manuscript, especially when involving very specific terms, whether in the medical or psychometric fields, that may be unfamiliar to the reader

The Authors: In the revised version of the manuscript we tried to explain the technical issues in more details which could be helpful for the Readers less familiar with the concepts.

Round 2

Reviewer 2 Report

Comments and Suggestions for Authors

Your research is thorough and the manuscript frames the need for population-specific classification benchmarks. Your findings and implications are appropriately described.  My only suggestion is very minor.

Section 1.2.  Consider breaking the single paragraph into two paragraphs.  Perhaps the 2nd paragraph could start with the sentence, "Although, Natoli et al. [34] show the measurement invariance of the LPFS-BF 2.0 across tested countries . . ."
(note misspelling 'stablished' line 151).

While there may be other similar revisions to consider, the manuscript is well done.